# On the logarithmic bipartite fidelity of the open XXZ spin chain at $\Delta = -1/2$

Christian Hagendorf[1][*] and Gilles Parez[2][†]

**1** Université catholique de Louvain, Institut de Recherche en Mathématique et Physique
Chemin du Cyclotron 2, 1348 Louvain-la-Neuve, Belgium
**2** Centre de Recherches Mathématiques (CRM), Université de Montréal,
P.O. Box 6128, Centre-ville Station,
Montréal (Québec), H3C 3J7, Canada

★ christian.hagendorf@uclouvain.be, † gilles.parez@umontreal.ca

## Abstract

The open XXZ spin chain with the anisotropy $\Delta = -\frac{1}{2}$ and a one-parameter family of diagonal boundary fields is studied at finite length. A determinant formula for an overlap involving the spin chain's ground-state vectors for different lengths is found. The overlap allows one to obtain an exact finite-size formula for the ground state's logarithmic bipartite fidelity. The leading terms of its asymptotic series for large chain lengths are evaluated. Their expressions confirm the predictions of conformal field theory for the fidelity.


# 1   Introduction and results

The study of entanglement in extended quantum many-body systems stimulates intense theoretical and experimental efforts in modern condensed matter physics [1]. This interest comes from the observation that entanglement measures, such as the celebrated entanglement entropy, are powerful tools to detect quantum phase transitions [2–4]. Moreover, they often characterise the quantum field theory that describes a system at (or in the vicinity of) the transition point [5, 6].

In this article, we investigate the so-called *logarithmic bipartite fidelity* (LBF). Dubail and Stéphan introduced this quantity to characterise the entanglement properties of the ground state of an extended quantum system [7]. Similarly to the entanglement entropy, it possesses an area-law property off criticality [8]. The violation of this area law at quantum criticality provides valuable insights into the system's description by conformal field theory (CFT) at the transition point [9].

The system that we focus on is the open XXZ chain with diagonal magnetic fields. For $N \geqslant 1$ sites, its Hamiltonian is given by[1]

$$H = -\frac{1}{2}\sum_{i=1}^{N-1}\left(\sigma_i^x \sigma_{i+1}^x + \sigma_i^y \sigma_{i+1}^y + \Delta \sigma_i^z \sigma_{i+1}^z\right) + p\sigma_1^z + \bar{p}\sigma_N^z. \tag{1.1}$$

Here, $\sigma_i^x$, $\sigma_i^y$, and $\sigma_i^z$ are the standard Pauli matrices acting on the site $1 \leqslant i \leqslant N$ of the chain. (We give a precise definition of the notations used in this introduction in Section 2.) Moreover, $\Delta$ is the anisotropy parameter, and $p, \bar{p}$ are the boundary magnetic fields at the left and the right ends of the chain. In the following, we focus on the case where these parameters are chosen in such a way that the Hamiltonian's ground-state eigenvalue is non-degenerate for any $N \geqslant 1$. Let $|\psi_N\rangle$ be a vector that spans the corresponding ground-state eigenspace. We refer to it as the ground-state vector. The LBF is defined in terms of this vector as

$$\mathcal{F}_{N_1,N_2} = -\ln\left(\frac{\left|\langle \psi_N | \left(|\psi_{N_1}\rangle \otimes |\psi_{N_2}\rangle\right)\right|^2}{\|\psi_N\|^2 \|\psi_{N_1}\|^2 \|\psi_{N_2}\|^2}\right), \tag{1.2}$$

where $N_1, N_2 \geqslant 1$ are integers such that $N = N_1 + N_2$. Hence, it measures to what extent the ground-state vector of a chain of length $N$ resembles the tensor product of the ground-state vectors of smaller chains with lengths $N_1, N_2$.

Despite the simplicity of its definition, exact finite-size expressions for the LBF are scarce in the literature, even for the well-studied XXZ Hamiltonian (1.1). For $\Delta = 0$, exact results are available for vanishing boundary fields $p = \bar{p} = 0$ [9] and Pasquier-Saleur boundary conditions $p = -\bar{p} = \mathrm{i}$ [10]. The key to finding these results are free-fermion methods. The point $\Delta = -\frac{1}{2}$ with $p = \bar{p} = -\frac{1}{4}$ provides another example for which the finite-size LBF is explicitly known [11, 12]. For this choice of the parameters, the spin-chain Hamiltonian (1.1) possesses an exact lattice supersymmetry and its ground-state vector has a rich combinatorial structure. Both

---

[1]For $N = 1$, the Hamiltonian is $H = (p + \bar{p})\sigma^z$ by convention.

these properties allow one to find the LBF for chains of finite lengths. Several generalisations to periodic and mixed boundary conditions have recently been investigated for $\Delta = 0$ and $\Delta = -\frac{1}{2}$, too [13,14]. Beyond the XXZ chain, an explicit expression for the LBF is known for the staggered supersymmetric $M_1$ model of strongly-interacting fermions with open boundary conditions [15].

**The ground state**

This article presents a new exact lattice calculation of the LBF for the open XXZ chain in finite size for a case that can be treated neither by free fermions nor by supersymmetry methods. We define this case through the choice

$$\Delta = -\frac{1}{2}, \quad p = \frac{1}{2}\left(\frac{1}{2} - x\right), \quad \bar{p} = \frac{1}{2}\left(\frac{1}{2} - \frac{1}{x}\right), \tag{1.3}$$

where $x$ is a real parameter. For this choice, the spin-chain Hamiltonian possesses the remarkably simple eigenvalue [16–18]

$$E_0 = -\frac{3N-1}{4} - \frac{(1-x)^2}{2x}. \tag{1.4}$$

At the supersymmetric point $x = 1$, it is the Hamiltonian's non-degenerate ground-state eigenvalue [11]. By continuity, this property still holds in the vicinity of the supersymmetric point. Moreover, numerical data supports the conjecture that $E_0$ is the non-degenerate ground-state value for all $x > 0$. In the following, we focus on $x > 0$ and assume that this conjecture holds.

The ground-state vector $|\psi_N\rangle$ spanning the eigenspace of $E_0$ was explicitly constructed in [12]. It has magnetisation 0 if $N$ is even, and $+\frac{1}{2}$ if $N$ is odd. By convention, its normalisation is fixed through the choice

$$(\psi_N)\underbrace{\downarrow \cdots \downarrow}_{n}\underbrace{\uparrow \cdots \uparrow}_{\bar{n}} = 1, \tag{1.5}$$

where $n = \lfloor N/2 \rfloor$, $\bar{n} = \lceil N/2 \rceil$. With this normalisation, the vector's components are polynomials in $x$ with integer coefficients. Some of them display remarkable relations with the enumerative combinatorics of alternating sign matrices and plane partitions [19]. Similar relations have been thoroughly investigated for the XXZ spin chain at $\Delta = -\frac{1}{2}$ with (twisted) periodic boundary conditions [20–27].

**Finite-size results**

In the following, we exploit the rich combinatorial and analytical properties of the ground-state vector to find the LBF. To this end, we write it as

$$\mathcal{F}_{N_1,N_2} = -\ln\left(\frac{\mathcal{O}_{N_1,N_2}^2}{\mathcal{O}_{N_1,0}\mathcal{O}_{N_2,0},\mathcal{O}_{N_1+N_2,0}}\right), \tag{1.6}$$

where $\mathcal{O}_{N_1,N_2}$ is the overlap of the ground-state vector for a chain of length $N = N_1 + N_2$ with the tensor product of the ground-state vectors for two chains of lengths $N_1$ and $N_2$:

$$\mathcal{O}_{N_1,N_2} = \langle\psi_N|\left(|\psi_{N_1}\rangle \otimes |\psi_{N_2}\rangle\right). \tag{1.7}$$

We include the cases $N_1 = 0$ or $N_2 = 0$, where the overlap corresponds to the square norm:

$$\mathcal{O}_{0,N} = \mathcal{O}_{N,0} = \langle\psi_N|\psi_N\rangle. \tag{1.8}$$

It will also be convenient to define $\mathcal{O}_{0,0} = 1$. Clearly, we have $\mathcal{O}_{N_1,N_2} = 0$ if both $N_1$ and $N_2$ are odd, because of the magnetisation of the ground-state vectors in the definition of the overlap. In this case, $\mathcal{F}_{N_1,N_2}$ is ill-defined.

Our first main result provides an explicit non-trivial expression for $\mathcal{O}_{N_1,N_2}$, and thus for $\mathcal{F}_{N_1,N_2}$, in the case where at most one of the integers $N_1, N_2$ is odd. This expression was conjectured in [12]. We write it in terms of a determinant whose entries are quadratic polynomials in $x$ with simple integer coefficients and in terms of

$$\gamma_N = \begin{cases} A_V(N+1), & \text{for even } N, \\ N_8(N+1), & \text{for odd } N. \end{cases} \tag{1.9}$$

Here,

$$A_V(2k+1) = \frac{1}{2^k} \prod_{i=1}^{k} \frac{(6i-2)!(2i-1)!}{(4i-2)!(4i-1)!}, \quad N_8(2k) = \prod_{i=0}^{k-1} \frac{(3i+1)(6i)!(2i)!}{(4i)!(4i+1)!}, \tag{1.10}$$

are the number of vertically-symmetric alternating sign matrices of size $(2k+1) \times (2k+1)$ and the number of cyclically-symmetric transpose complement plane partitions in a cube of size $2k \times 2k \times 2k$, respectively [19].

**Theorem 1.1.** *For all $N_1, N_2 \geqslant 0$, let $n = \lfloor (N_1 + N_2)/2 \rfloor$. For even $N_1, N_2$, we have*

$$\mathcal{O}_{N_1,N_2} = \gamma_{N_1} \gamma_{N_2} \det_{i,j=1}^{n} \left( (x-1)^2 \binom{i+j-2}{2j-i-1} + x \binom{i+j}{2j-i} \right). \tag{1.11}$$

*For even $N_1$ and odd $N_2$, or odd $N_1$ and even $N_2$, we have*

$$\mathcal{O}_{N_1,N_2} = \gamma_{N_1} \gamma_{N_2} \det_{i,j=1}^{n} \left( (x-1)^2 \binom{i+j-1}{2j-i} + x \binom{i+j+1}{2j-i+1} \right). \tag{1.12}$$

We prove this theorem in Section 3. To this end, we utilise a well-established technique that consists of deriving exact finite-size results for the XXZ spin chain at $\Delta = -\frac{1}{2}$ from polynomial solutions of the quantum Knizhnik-Zamolodchikov equations associated to the six-vertex model [25,28]. In the present case, we work with Laurent-polynomial solutions of a version of these equations with boundaries. We introduce an overlap for these solutions that generalises the spin-chain overlap $\mathcal{O}_{N_1,N_2}$. The properties of the generalised overlap as a Laurent polynomial determine it uniquely. Exploiting this uniqueness property and applying a standard induction argument, we prove that the generalised overlap can be written as a product of so-called symplectic characters. The expressions of Theorem 1.1 follow from a specialisation of these symplectic characters.

**Asymptotic series**

The asymptotic behaviour of $\mathcal{F}_{N_1,N_2}$ for large $N_1, N_2$ is of particular interest. Indeed, Dubail and Stéphan argued that for one-dimensional quantum critical systems, the coefficients of the leading terms of its asymptotic series possess universal properties, which are predicted by CFT [7,9]. The asymptotic behaviour is the subject of our second main result. To state it, we use the parameterisation [29]

$$x = \frac{\sin(\pi(r+1)/3)}{\sin(\pi r/3)}. \tag{1.13}$$

In this parameterisation, each $x > 0$ can be obtained from a unique $0 < r < 2$. Furthermore, it will be useful to define the abbreviations

$$D = \frac{2}{\Gamma(1/3)} \sqrt{\frac{\pi}{3}} \frac{\sin(2\pi(r-1)/3)}{\sin(\pi(r-1)/2)}, \tag{1.14}$$

and

$$E = 13 - 14\sin^2\left(\frac{\pi(r-1)}{2}\right), \quad \bar{E} = 11 - 10\sin^2\left(\frac{\pi(r-1)}{2}\right). \qquad (1.15)$$

**Theorem 1.2.** *Let $0 < r < 2$, $N = N_1 + N_2$ and $\xi = N_1/N$. As $N_1, N_2 \to \infty$ (such that $\xi$ converges in $]0, 1[$), the LBF is asymptotically given by*

$$\mathcal{F}_{N_1,N_2} = \frac{1}{6}\ln N + \frac{1}{6}\ln(\xi(1-\xi)) - \ln D + \frac{E}{72}\left(\frac{1}{\xi} + \frac{1}{1-\xi} - 1\right)N^{-1} + o(N^{-1}), \qquad (1.16)$$

*for even $N_1, N_2$. For even $N_1$ and odd $N_2$, it is asymptotically given by*

$$\mathcal{F}_{N_1,N_2} = \frac{1}{6}\ln N + \frac{1}{6}\ln\left(\frac{\xi}{1-\xi}\right) - \ln D + \frac{1}{72}\left(\frac{E}{\xi} + \bar{E}\left(1 - \frac{1}{1-\xi}\right)\right)N^{-1} + o(N^{-1}). \qquad (1.17)$$

*For odd $N_1$ and even $N_2$, the asymptotic series follows from* (1.17) *with $\xi$ replaced by $1-\xi$.*

The proof of this theorem uses the asymptotic properties of symplectic characters, obtained by Gorin and Panova [30]. We provide this proof in Section 4 and argue that the leading terms of the asymptotic series up to $O(1)$ match the CFT prediction.

In Figure 1, we compare the leading terms of the asymptotic series and the finite-size LBF for $x = \frac{1}{2}$ and $N = 72, 73$ graphically. The inclusion of the $O(N^{-1})$-term leads to an approximation of the exact results by the truncated asymptotic series with a satisfactory precision.

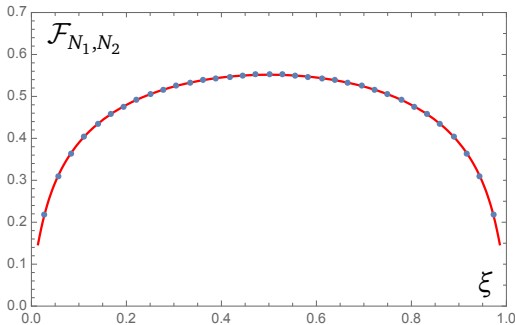 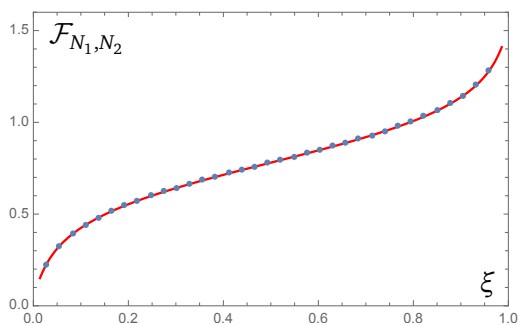

Figure 1: The figure shows a comparison of the LBF with $x = \frac{1}{2}$ and its approximation by the leading terms of its asymptotic series. We have $N = 72$ with even $N_1, N_2$ on the left panel, and $N = 73$ with even $N_1$ and odd $N_2$ on the right panel. The dots correspond to the exact finite-size values as a function of $\xi = N_1/N$. The solid lines represent the asymptotic series, truncated after the $O(N^{-1})$-term and plotted as a function of $0 < \xi < 1$. It approximates the exact values with a precision of the order of $10^{-3}$.

**Outline**

This article is organised as follows. In Section 2, we define and analyse a multi-parameter generalisation of the spin-chain overlap with the help of a solution to the boundary quantum Knizhnik-Zamolodchikov equations related to the six-vertex model. For a particular choice of the parameters, we compute the overlap in terms of symplectic characters. The purpose of Section 3 is to analyse the so-called homogeneous limit of the generalised overlap. This limit allows us to recover the spin-chain overlap and prove Theorem 1.1. We prove Theorem 1.2 in Section 4, and use it to discuss the relation between the leading terms of the LBF's asymptotic series and CFT. Section 5 contains our conclusions and discusses several open problems.

# 2 An overlap for a solution of the boundary quantum Knizhnik-Zamolodchikov equations

In this section, we consider a generalisation of the spin-chain overlap. The purpose of Section 2.1 is to set our notations and conventions, recall a few elements of the theory of the six-vertex model, and provide a short summary of several properties of a solution to the boundary quantum Knizhnik-Zamolodchikov equations associated with the six-vertex model. In Section 2.2, we establish a reduction relation for this solution. We introduce and study the generalised overlap in Section 2.3. Using its properties, we find in Section 2.4 an explicit expression of this overlap at the so-called combinatorial point in terms of symplectic characters.

## 2.1 Notations

**Space of states and operators**

Throughout the following, $N \geqslant 1$ denotes an integer. The space of states of the spin chain of length $N$ is $V^N = V_1 \otimes V_2 \otimes \cdots \otimes V_N$, where $V_i = \mathbb{C}^2$ is the local space of states of the spin at site $1 \leqslant i \leqslant N$. This local space is spanned by the canonical basis vectors

$$|\uparrow\rangle = \begin{pmatrix} 1 \\ 0 \end{pmatrix}, \quad |\downarrow\rangle = \begin{pmatrix} 0 \\ 1 \end{pmatrix}. \tag{2.1}$$

The canonical basis vectors of $V^N$ are the tensor products $|\alpha_1 \alpha_2 \cdots \alpha_N\rangle = |\alpha_1\rangle \otimes |\alpha_2\rangle \otimes \cdots \otimes |\alpha_N\rangle$, where $\alpha_1, \ldots, \alpha_N \in \{\uparrow, \downarrow\}$. The components of a vector $|\psi\rangle \in V^N$ are the coefficients in its expansion along the basis vectors:

$$|\psi\rangle = \sum_{\alpha_1 \alpha_2 \cdots \alpha_N \in \{\uparrow,\downarrow\}^N} \psi_{\alpha_1 \cdots \alpha_N} |\alpha_1 \alpha_2 \cdots \alpha_N\rangle. \tag{2.2}$$

For each vector $|\psi\rangle \in V^N$, we define a co-vector $\langle\psi| \in (V^N)^*$ by transposition, $\langle\psi| = |\psi\rangle^t$. The action of a co-vector $\langle\phi| \in (V^N)^*$ on a vector $|\psi\rangle \in V^N$ is

$$\langle\phi|\psi\rangle = \sum_{\alpha_1 \alpha_2 \cdots \alpha_N \in \{\uparrow,\downarrow\}^N} \phi_{\alpha_1 \alpha_2 \cdots \alpha_N} \psi_{\alpha_1 \alpha_2 \cdots \alpha_N}. \tag{2.3}$$

We refer to it as the *overlap* between $|\phi\rangle$ and $|\psi\rangle$, which is different from the standard Hermitian scalar product in quantum mechanics. However, for vectors with real components, such as the spin chain's ground-state vector discussed in the introduction, it defines a real scalar product. For any such vector $|\psi\rangle$, the overlap $\langle\psi|\psi\rangle$ yields the square norm.

Let $A \in \text{End}(V^N)$ be a linear operator on the spin chain's space of states. Its action on the canonical basis can be encoded in a $2^N \times 2^N$ matrix. Since we only work with the canonical basis, we do not distinguish the operator from its matrix representation. Let $1 \leqslant M \leqslant N$ be an integer and $A \in \text{End}(V^M)$. For each sequence $1 \leqslant i_1 < i_2 < \cdots < i_M \leqslant N$, we use the tensor-leg notation $A_{i_1,\ldots,i_M}$ for the canonical embedding of $A$ into $\text{End}(V^N)$. For example, if $M = 1$, we have

$$A_i = \underbrace{1 \otimes \cdots \otimes 1}_{i-1} \otimes A \otimes \underbrace{1 \otimes \cdots \otimes 1}_{N-i}, \tag{2.4}$$

where $1$ denotes the identity operator on $\mathbb{C}^2$. In particular, we used this notation for the standard Pauli matrices

$$\sigma^x = \begin{pmatrix} 0 & 1 \\ 1 & 0 \end{pmatrix}, \quad \sigma^y = \begin{pmatrix} 0 & -i \\ i & 0 \end{pmatrix}, \quad \sigma^z = \begin{pmatrix} 1 & 0 \\ 0 & -1 \end{pmatrix}, \tag{2.5}$$

in the definition of the spin-chain Hamiltonian (1.1).

The Hamiltonian commutes with the magnetisation operator $\mathcal{M}$. On $V^N$, it is given by

$$\mathcal{M} = \frac{1}{2} \sum_{i=1}^{N} \sigma_i^z. \tag{2.6}$$

The commutation implies that $H$ and $\mathcal{M}$ can be simultaneously diagonalised. The spectrum of $\mathcal{M}$ on $V^N$ is $\{-N/2, -N/2+1, \ldots, N/2\}$. We say that $|\psi\rangle \in V^N$ has magnetisation $\mu$ if it is an eigenvector of $\mathcal{M}$ with eigenvalue $\mu$.

### The six-vertex model: $\check{R}$-matrix and $K$-matrix

The $\check{R}$-matrix of the six-vertex model is an operator $\check{R}(z) \in \mathrm{End}(V^2)$ that depends on a spectral parameter $z$. It acts on the canonical basis of $V^2$ as the matrix

$$\check{R}(z) = \begin{pmatrix} a(z) & 0 & 0 & 0 \\ 0 & c(z) & b(z) & 0 \\ 0 & b(z) & c(z) & 0 \\ 0 & 0 & 0 & a(z) \end{pmatrix}. \tag{2.7}$$

The non-zero entries of this matrix are the vertex weights of the six-vertex model [31]. We parameterise them in terms of $z$ and the crossing parameter $q$, according to

$$a(z) = [qz]/[q/z], \quad b(z) = [z]/[q/z], \quad c(z) = [q]/[q/z]. \tag{2.8}$$

Here, we used the abbreviation

$$[z] = z - z^{-1}. \tag{2.9}$$

In this parameterisation, the $\check{R}$-matrix obeys the (braid) Yang-Baxter equation: On $V^3$, we have

$$\check{R}_{1,2}(z/w)\check{R}_{2,3}(z)\check{R}_{1,2}(w) = \check{R}_{2,3}(w)\check{R}_{1,2}(z)\check{R}_{2,3}(z/w). \tag{2.10}$$

In addition to the $\check{R}$-matrix, we consider a diagonal $K$-matrix for the six-vertex model. It acts on the canonical basis of $V^1$ as the matrix

$$K(z) = \begin{pmatrix} 1 & 0 \\ 0 & [\beta z]/[\beta/z] \end{pmatrix}. \tag{2.11}$$

We sometimes write $K(z) = K(z; \beta)$ to stress the dependence on $\beta$. The $K$-matrix is a solution of the boundary Yang-Baxter equation of the six-vertex model [32]. On $V^2$, it can be written as

$$\check{R}_{1,2}(z/w)K_1(z)\check{R}_{1,2}(zw)K_1(w) = K_1(w)\check{R}_{1,2}(zw)K_1(z)\check{R}_{1,2}(z/w). \tag{2.12}$$

The expression (2.11) is one of the simplest solutions to this equation [33]. The general solution was found in [34, 35].

### A solution to the boundary quantum Knizhnik-Zamolodchikov equations

The building block of our generalisation of the spin-chain overlap is a vector $|\Psi_N\rangle \in V^N$. It was constructed and studied in [12]. Here, we summarise the properties of this vector that are central to our investigation.

The vector's magnetisation is $\mu = (\bar{n} - n)/2$, where

$$n = \lfloor N/2 \rfloor, \quad \bar{n} = \lceil N/2 \rceil. \tag{2.13}$$

The components of the vector that do not vanish trivially have explicit expressions in terms of multiple contour integrals, and depend on the complex numbers $z_1, \ldots, z_N$ and $q, \beta$. We often write $|\Psi_N\rangle = |\Psi_N(z_1, \ldots, z_N)\rangle$ to stress the dependence on $z_1, \ldots, z_N$. A crucial property of the vector is that it obeys the so-called *exchange relations* [12, Proposition 3.2]: For $N \geqslant 2$ and each $1 \leqslant i \leqslant N - 1$, we have

$$\check{R}_{i,i+1}(z_i/z_{i+1})|\Psi_N(\ldots, z_i, z_{i+1}, \ldots)\rangle = |\Psi_N(\ldots, z_{i+1}, z_i, \ldots)\rangle. \tag{2.14}$$

These relations are compatible thanks to the Yang-Baxter equation (2.10). They lead to a system of linear equations for the vector's components. These equations imply that all its non-identically vanishing components can be inferred from the sole knowledge of the following factorised component:

$$(\Psi_N)\underbrace{\downarrow \cdots \downarrow}_{n} \underbrace{\uparrow \cdots \uparrow}_{\bar{n}} = \prod_{i=1}^{n} [\beta z_i] \prod_{1 \leqslant i < j \leqslant n} [q z_j / z_i][q z_i z_j] \prod_{n+1 \leqslant i < j \leqslant N} [q z_j / z_i][q^2 z_i z_j]. \tag{2.15}$$

This expression follows from the component's multiple contour-integral formula [12, Proposition 3.1]. In addition to the exchange relations, the vector $|\Psi_N\rangle$ also obeys two *reflection relations* [12, Proposition 3.7]. They express the local action of the $K$-matrix on the vector:

$$K_1(z_1^{-1}; \beta)|\Psi_N(z_1, \ldots, z_N)\rangle = |\Psi_N(z_1^{-1}, \ldots, z_N)\rangle, \tag{2.16a}$$

$$K_N(s z_N; s q^{-1} \beta^{-1})|\Psi_N(z_1, \ldots, z_N)\rangle = |\Psi_N(z_1, \ldots, s^{-2} z_N^{-1})\rangle. \tag{2.16b}$$

Here, the parameter $s$ satisfies $s^2 = q^3$. (It is possible to work with the less restrictive condition $s^4 = q^6$ [12], but this is not necessary for the following.) The exchange and reflection relations are compatible thanks to the boundary Yang-Baxter equation (2.12). Moreover, the combination of (2.14) and (2.16) implies that $|\Psi_N\rangle$ is a solution to the boundary quantum Knizhnik-Zamolodchikov equations [36].

Using the exchange relations (2.14) and the simple component (2.15), it is straightforward to construct the components of the vector $|\Psi_N\rangle$ that do not vanish identically for small values of $N$. For $N = 1$, the only such component is

$$(\Psi_1)_\uparrow = 1. \tag{2.17}$$

For $N = 2$, we find

$$(\Psi_2)_{\downarrow\uparrow} = [\beta z_1], \quad (\Psi_2)_{\uparrow\downarrow} = -[q \beta z_2]. \tag{2.18}$$

For $N = 3$, we have

$$(\Psi_3)_{\downarrow\uparrow\uparrow} = [\beta z_1][q z_3 / z_2][q^2 z_2 z_3], \quad (\Psi_3)_{\uparrow\uparrow\downarrow} = [q \beta z_3][q z_2 / z_1][q z_1 z_2], \tag{2.19}$$

and

$$(\Psi_3)_{\uparrow\downarrow\uparrow} = \frac{[q][\beta z_1][q z_3 / z_2][q^2 z_2 z_3] - [\beta z_2][q z_2 / z_1][q z_3 / z_1][q^2 z_1 z_3]}{[z_2 / z_1]}. \tag{2.20}$$

Despite its appearance, this last component actually is a Laurent polynomial in $z_1, z_2, z_3$. This property generalises to arbitrary $N$: Each component of $|\Psi_N\rangle$ is a Laurent polynomial in $z_i$ for each $1 \leqslant i \leqslant N$ [12, Propositions 3.11, 3.12].

## 2.2 Reduction relations

We now show that the vector $|\Psi_N\rangle$ simplifies if $z_{i+1} = q^{-1}z_i$ for some $1 \leqslant i \leqslant N-1$. An example of such a simplification occurs for $N = 2$ and $i = 1$, where the specialisation essentially yields a singlet $|\zeta\rangle = |\uparrow\downarrow\rangle - |\downarrow\uparrow\rangle$:

$$|\Psi_2(z_1, z_2 = q^{-1}z_1)\rangle = -[\beta z_1]|\zeta\rangle. \tag{2.21}$$

For $N \geqslant 3$, a similar phenomenon occurs. Upon specialisation, the vector $|\Psi_N\rangle$ simplifies to an expression involving $|\zeta\rangle$ and $|\Psi_{N-2}\rangle$. We call the resulting relation between $|\Psi_N\rangle$ and $|\Psi_{N-2}\rangle$ a *reduction relation*.

To write the reduction relation for $N \geqslant 3$, we introduce the linear mappings $\Xi_N^i : V^{N-2} \to V^N$, $1 \leqslant i \leqslant N-1$. They are defined by their action on the basis vectors of $V^{N-2}$: For $\alpha_1, \ldots, \alpha_{N-2} \in \{\uparrow, \downarrow\}$, we have

$$\Xi_N^i |\alpha_1 \cdots \alpha_{i-1} \alpha_i \cdots \alpha_{N-2}\rangle = |\alpha_1 \cdots \alpha_{i-1}\rangle \otimes |\zeta\rangle \otimes |\alpha_i \cdots \alpha_{N-2}\rangle. \tag{2.22}$$

We note that $\Xi_N^i$ is injective. In the following, we need to evaluate the action of several $\check{R}$-matrices on $\Xi_N^i$. To this end, it is convenient to define the abbreviations

$$\mathbb{R}_i(z, w) = \check{R}_{i,i+1}(z)\check{R}_{i-1,i}(w), \quad \bar{\mathbb{R}}_i(z, w) = \check{R}_{i-1,i}(z)\check{R}_{i,i+1}(w). \tag{2.23}$$

**Lemma 2.1.** *For $2 \leqslant i \leqslant N-1$, we have*

$$\mathbb{R}_i(qz, z)\Xi_N^i = -\frac{[q^2 z]}{[q/z]}\Xi_N^{i-1}, \quad \bar{\mathbb{R}}_i(qz, z)\Xi_N^{i-1} = -\frac{[q^2 z]}{[q/z]}\Xi_N^i. \tag{2.24}$$

*Proof.* Let $P$ be the linear operator on $V^2$ defined through the action $P|\alpha_1\alpha_2\rangle = |\alpha_2\alpha_1\rangle$, for all $\alpha_1, \alpha_2 \in \{\uparrow, \downarrow\}$. It is straightforward to check the identity

$$\mathbb{R}_i(qz, z)\check{R}_{i,i+1}(q^{-1}) = \frac{[q^2 z]}{[q/z]}P_{i-1,i+1}\check{R}_{i,i+1}(q^{-1}). \tag{2.25}$$

We apply both sides of this identity to $\Xi_N^i$ and use $\check{R}_{i,i+1}(q^{-1})\Xi_N^i = (2[q]/[q^2])\Xi_N^i$. This application leads to

$$\mathbb{R}_i(qz, z)\Xi_N^i = \frac{[q^2 z]}{[q/z]}P_{i-1,i+1}\Xi_N^i. \tag{2.26}$$

Using (2.22), we have $P_{i-1,i+1}\Xi_N^i = -\Xi_N^{i-1}$ and obtain, thus, the first equality of (2.24). The proof of the second equality is similar. $\qquad\square$

We now use Lemma 2.1 to find the reduction relations for $N \geqslant 3$.

**Proposition 2.2.** *For $N \geqslant 3$ and $1 \leqslant i \leqslant N-1$, we have*

$$|\Psi_N(\ldots, z_{i-1}, z_i, z_{i+1} = q^{-1}z_i, z_{i+2}, \ldots)\rangle = (-1)^{n+i+1}[\beta z_i] \prod_{j=1}^{i-1}[qz_i/z_j][qz_i z_j]$$

$$\times \prod_{j=i+2}^{N}[q^2 z_j/z_i][qz_i z_j]\, \Xi_N^i |\Psi_{N-2}(\ldots, z_{i-1}, z_{i+2}, \ldots)\rangle, \tag{2.27}$$

*where n is defined in (2.13).*

*Proof.* The proof consists of two steps. In step 1, we prove (2.27) for $i = n$. In step 2, we extend it to all $1 \leqslant i \leqslant N - 1$.

*Step 1:* For $i = n$, the exchange relations (2.14) imply

$$|\Psi_N(\ldots, z_n, z_{n+1} = q^{-1}z_n, \ldots)\rangle = \check{R}_{n,n+1}(q^{-1})|\Psi_N(\ldots, q^{-1}z_n, z_n, \ldots)\rangle. \tag{2.28}$$

The $\check{R}$-matrix on the right-hand side is given by $\check{R}(q^{-1}) = ([q]/[q^2])|\zeta\rangle\langle\zeta|$. Hence, there is a vector $|\Phi\rangle = |\Phi(z_n; z_1, \ldots, z_{n-1}, z_{n+2}, \ldots, z_N)\rangle \in V^{N-2}$ of magnetisation $(\bar{n} - n)/2$, where $\bar{n}$ is defined in (2.13), such that[2]

$$|\Psi_N(\ldots, z_{n-1}, z_n, z_{n+1} = q^{-1}z_n, z_{n+1} \ldots)\rangle$$
$$= -[\beta z_n]\prod_{j=1}^{n-1}[qz_n/z_j][qz_nz_j]\prod_{j=n+2}^{N}[q^2z_j/z_n][qz_nz_j]\Xi_N^n|\Phi(z_n; \ldots, z_{n-1}, z_{n+2}, \ldots)\rangle. \tag{2.29}$$

We note that $|\Phi\rangle$ is unique because $\Xi_N^n$ is injective.

To find $|\Phi\rangle$, we establish two properties of this vector. First, it has the simple component

$$\Phi_{\underbrace{\downarrow\cdots\downarrow}_{n-1}\underbrace{\uparrow\cdots\uparrow}_{\bar{n}-1}}(z_n; z_1, \ldots, z_{n-1}, z_{n+2}, \ldots, z_N) = (\Psi_{N-2})_{\underbrace{\downarrow\cdots\downarrow}_{n-1}\underbrace{\uparrow\cdots\uparrow}_{\bar{n}-1}}(z_1, \ldots, z_{n-1}, z_{n+2}, \ldots, z_N). \tag{2.30}$$

This equality straightforwardly follows from (2.15) and (2.29). Second, $|\Phi\rangle$ obeys the same exchange relations as $|\Psi_{N-2}(z_1, \ldots, z_{n-1}, z_{n+2}, \ldots, z_N)\rangle$. Indeed, combining (2.14) with (2.29), we find

$$\check{R}_{i,i+1}\left(\frac{z_i}{z_{i+1}}\right)|\Phi(\ldots, z_i, z_{i+1}, \ldots)\rangle = |\Phi(\ldots, z_{i+1}, z_i, \ldots)\rangle, \quad 1 \leqslant i \leqslant n-2, \tag{2.31}$$

$$\check{R}_{i,i+1}\left(\frac{z_{i+2}}{z_{i+3}}\right)|\Phi(\ldots, z_{i+2}, z_{i+3}, \ldots)\rangle = |\Phi(\ldots, z_{i+3}, z_{i+2}, \ldots)\rangle, \quad n \leqslant i \leqslant N-2. \tag{2.32}$$

Moreover, we claim that the following exchange relation holds:

$$\check{R}_{n-1,n}(z_{n-1}/z_{n+2})|\Phi(\ldots, z_{n-1}, z_{n+2}, \ldots)\rangle = |\Phi(\ldots, z_{n+2}, z_{n-1}, \ldots)\rangle. \tag{2.33}$$

To see this, we use (2.14) to write

$$\mathbb{R}_{n+1}\left(\frac{z_{n-1}}{z_{n+1}}, \frac{z_{n-1}}{z_n}\right)\check{R}_{n-1,n}\left(\frac{z_{n-1}}{z_{n+2}}\right)\bar{\mathbb{R}}_{n+1}\left(\frac{z_n}{z_{n+2}}, \frac{z_{n+1}}{z_{n+2}}\right)$$
$$\times |\Psi_N(\ldots, z_{n-1}, z_n, z_{n+1}, z_{n+2}, \ldots)\rangle = |\Psi_N(\ldots, z_{n+2}, z_n, z_{n+1}, z_{n-1}, \ldots)\rangle. \tag{2.34}$$

We now set $z_{n+1} = q^{-1}z_n$. Using (2.29), we find

$$\mathbb{R}_{n+1}\left(\frac{qz_{n-1}}{z_n}, \frac{z_{n-1}}{z_n}\right)\check{R}_{n-1,n}\left(\frac{z_{n-1}}{z_{n+2}}\right)\bar{\mathbb{R}}_{n+1}\left(\frac{z_n}{z_{n+2}}, \frac{q^{-1}z_n}{z_{n+2}}\right)\Xi_N^n|\Phi(\ldots, z_{n-1}, z_{n+2}, \ldots)\rangle$$
$$= \frac{[qz_n/z_{n+2}][q^2z_{n-1}/z_n]}{[qz_n/z_{n-1}][q^2z_{n+2}/z_n]}\Xi_N^n|\Phi(\ldots, z_{n+2}, z_{n-1}, \ldots)\rangle. \tag{2.35}$$

Next, we apply Lemma 2.1 and simplify this equality to

$$\mathbb{R}_{n+1}\left(\frac{qz_{n-1}}{z_n}, \frac{z_{n-1}}{z_n}\right)\check{R}_{n-1,n}\left(\frac{z_{n-1}}{z_{n+2}}\right)\Xi_N^{n+1}|\Phi(\ldots, z_{n-1}, z_{n+2}, \ldots)\rangle$$
$$= -\frac{[q^2z_{n-1}/z_n]}{[qz_n/z_{n-1}]}\Xi_N^n|\Phi(\ldots, z_{n+2}, z_{n-1}, \ldots)\rangle. \tag{2.36}$$

---

[2]Including the pre-factor on the right-hand side of the equality turns out to be convenient in the following.

The operators $\check{R}_{n-1,n}(z_{n-1}/z_{n+2})$ and $\Xi_N^{n+1}$ on the right-hand side of this equality commute, which straightforwardly follows from (2.22). Moreover, the action of $\mathbb{R}_{n+1}(qz_{n-1}/z_n, z_{n-1}/z_n)$ onto $\Xi_N^{n+1}$ follows from Lemma 2.1. We combine these two observations and find

$$\Xi_N^n \check{R}_{n-1,n}\left(\frac{z_{n-1}}{z_{n+2}}\right)|\Phi(\dots, z_{n-1}, z_{n+2}, \dots)\rangle = \Xi_N^n|\Phi(\dots, z_{n+2}, z_{n-1}, \dots)\rangle. \tag{2.37}$$

Since $\Xi_N^n$ is injective, we conclude that (2.33) holds.

In conclusion, $|\Phi(z_n; z_1, \dots, z_{n-1}, z_{n+2}, \dots, z_N)\rangle$ and $|\Psi_{N-2}(z_1, \dots, z_{n-1}, z_{n+2}, \dots, z_N)$ have the same magnetisation, one common non-vanishing component, and obey the same exchange relations. It follows from [12, Proposition 3.5] that the two vectors are equal. This proves (2.27) for $i = n$.

*Step 2:* Let us suppose that $i < n$. We use (2.14) to write

$$|\Psi_N(\dots, z_i, z_{i+1}, \dots)\rangle$$
$$= \prod_{j=i+2}^{n+1} \mathbb{R}_{j-1}(z_j/z_{i+1}, z_j/z_i)|\Psi_N(\dots, z_{i-1}, z_{i+2}, \dots, z_{n+1}, z_i, z_{i+1}, z_{n+2}, \dots)\rangle. \tag{2.38}$$

The $n$-th and $(n+1)$-th arguments of the vector on the right-hand side are $z_i$ and $z_{i+1}$, respectively. We now set $z_{i+1} = q^{-1}z_i$, and apply the reduction relation obtained in step 1. It yields

$$|\Psi_N(\dots, z_i, z_{i+1} = q^{-1}z_i, \dots)\rangle = -[\beta z_i] \prod_{j=1}^{i-1}[qz_i/z_j][qz_iz_j] \prod_{j=i+2}^{n+1}[qz_i/z_j][qz_iz_j]$$
$$\times \prod_{j=n+2}^{N}[q^2z_j/z_i][qz_iz_j] \prod_{j=i+2}^{n+1} \mathbb{R}_{j-1}(qz_j/z_i, z_j/z_i)\Xi_N^n|\Psi_{N-2}(\dots, z_{i-1}, z_{i+2}, \dots)\rangle. \tag{2.39}$$

By Lemma 2.1, we have

$$\prod_{j=i+2}^{n+1} \mathbb{R}_{j-1}(qz_j/z_i, z_j/z_i)\Xi_N^n = (-1)^{n+i}\left(\prod_{j=i+2}^{n+1} \frac{[q^2z_j/z_i]}{[qz_i/z_j]}\right)\Xi_N^i, \tag{2.40}$$

for each $i + 1 \leqslant j \leqslant n$. Using this relation in (2.39), we obtain (2.27) for $i < n$. The proof for $i > n$ is similar. $\qquad\square$

## 2.3 The overlap

Throughout this section, $N_1, N_2 \geqslant 0$ are integers and $N = N_1 + N_2$, unless stated otherwise. For each $N_1, N_2 \geqslant 1$, we define the generalised overlap

$$\Omega_{N_1,N_2} = \langle\Psi_N(z_1^{-1}, \dots, z_{N_1}^{-1}, q^{-3}z_{N_1+1}^{-1}, \dots, q^{-3}z_N^{-1})|$$
$$\times \left(|\Psi_{N_1}(z_1, \dots, z_{N_1})\rangle \otimes |\Psi_{N_2}(z_{N_1+1}, \dots, z_N)\rangle\right). \tag{2.41}$$

By convention, if $N_1 = 0$ or $N_2 = 0$, we replace the corresponding tensor factor by the scalar factor 1. This convention implies

$$\Omega_{0,N} = \langle\Psi_N(q^{-3}z_1^{-1}, \dots, q^{-3}z_N^{-1})|\Psi_N(z_1, \dots, z_N)\rangle, \tag{2.42}$$
$$\Omega_{N,0} = \langle\Psi_N(z_1^{-1}, \dots, z_N^{-1})|\Psi_N(z_1, \dots, z_N)\rangle, \tag{2.43}$$

and $\Omega_{0,0} = 1$.

By construction, $\Omega_{N_1,N_2}$ is a Laurent polynomial in $z_1, \dots, z_N$. It identically vanishes if both $N_1$ and $N_2$ are odd, because of the magnetisation of the vectors that we use to compute the overlap. For the other parities of $N_1, N_2$, we expect $\Omega_{N_1,N_2}$ to be a non-trivial function of $z_1, \dots, z_N$. In the following, we characterise it as a function of these variables. We often write $\Omega_{N_1,N_2} = \Omega_{N_1,N_2}(z_1, \dots, z_N)$ to stress its dependence on them.

**Symmetry**

**Lemma 2.3.** *We have*

$$\Omega_{N_1,N_2}(\dots, -z_i, \dots) = \Omega_{N_1,N_2}(\dots, z_i, \dots), \tag{2.44}$$

*for each* $1 \leqslant i \leqslant N$.

*Proof.* The proof straightforwardly follows from the relation

$$|\Psi_N(\dots, -z_i, \dots)\rangle = \sigma_i^z |\Psi_N(\dots, z_i, \dots)\rangle, \tag{2.45}$$

which holds for all $N \geqslant 1$ and is an immediate consequence of [12, Propositions 3.11 and 3.12]. $\qquad\square$

**Lemma 2.4.** *For* $N_1 \geqslant 2$, $\Omega_{N_1,N_2}$ *is a symmetric function of* $z_1, \dots, z_{N_1}$. *Likewise, for* $N_2 \geqslant 2$, *it is a symmetric function of* $z_{N_1+1}, \dots, z_N$.

*Proof.* First, we suppose that $N_1 \geqslant 2$ and consider an integer $1 \leqslant i \leqslant N_1 - 1$. We use the unitarity relation $\check{R}_{i,i+1}(z^{-1})\check{R}_{i,i+1}(z) = 1$ to write

$$\Omega_{N_1,N_2} = \langle \Psi_N(\dots, z_i^{-1}, z_{i+1}^{-1}, \dots) | \check{R}_{i,i+1}(z_{i+1}/z_i)$$
$$\times \left( \check{R}_{i,i+1}(z_i/z_{i+1}) |\Psi_{N_1}(\dots, z_i, z_{i+1}, \dots)\rangle \otimes |\Psi_{N_2}(\dots)\rangle \right). \tag{2.46}$$

We evaluate the action of the $\check{R}$-matrices on the vectors in the first and second line, respectively, with the help of the exchange relations (2.14) and the symmetry of the $\check{R}$-matrix. This evaluation leads to

$$\Omega_{N_1,N_2}(\dots, z_i, z_{i+1}, \dots) = \Omega_{N_1,N_2}(\dots, z_{i+1}, z_i, \dots), \tag{2.47}$$

which is enough to conclude that the overlap is symmetric in $z_1, \dots, z_{N_1}$. The proof of the symmetry in $z_{N_1+1}, \dots, z_N$ for $N_2 \geqslant 2$ follows the same steps. $\qquad\square$

**Lemma 2.5.** *For* $N_1 \geqslant 1$ *and each* $1 \leqslant i \leqslant N_1$, *we have*

$$\Omega_{N_1,N_2}(\dots, z_i^{-1}, \dots) = \Omega_{N_1,N_2}(\dots, z_i, \dots). \tag{2.48}$$

*For* $N_2 \geqslant 1$ *and each* $N_1 + 1 \leqslant i \leqslant N$, *we have*

$$\Omega_{N_1,N_2}(\dots, z_i^{-1}, \dots) = \Omega_{N_1,N_2}(\dots, q^{-3}z_i, \dots). \tag{2.49}$$

*Proof.* First, we suppose that $N_1 \geqslant 1$ and set $i = 1$. The reflection relations (2.16) allow us to write

$$\Omega_{N_1,N_2}(z_1^{-1}, \dots) = \langle \Psi_N(z_1, \dots) | \left( |\Psi_{N_1}(z_1^{-1}, \dots)\rangle \otimes |\Psi_{N_2}(\dots)\rangle \right) \tag{2.50}$$
$$= \langle \Psi_N(z_1^{-1}, \dots) | K_1(z_1; \beta) \left( K_1(z_1^{-1}; \beta) |\Psi_{N_1}(z_1, \dots)\rangle \otimes |\Psi_{N_2}(\dots)\rangle \right). \tag{2.51}$$

Using $K(z;\beta)K(z^{-1};\beta) = 1$, we obtain

$$\Omega_{N_1,N_2}(z_1^{-1},\dots) = \Omega_{N_1,N_2}(z_1,\dots),\tag{2.52}$$

which concludes the proof of (2.48) for $i = 1$. The generalisation to $2 \leqslant i \leqslant N_1$ follows from Lemma 2.4.

Second, we suppose that $N_2 \geqslant 1$ and set $i = N$. We use the reflection relations (2.16) and the relation $s^2 = q^3$ to find

$$\begin{aligned}
\Omega_{N_1,N_2}(\dots,z_N^{-1}) &= \langle\Psi_N(\dots,q^{-3}z_N)|\left(|\Psi_{N_1}(\dots)\rangle \otimes |\Psi_{N_2}(\dots,z_N^{-1})\rangle\right) &(2.53)\\
&= \langle\Psi_N(\dots,z_N^{-1})|K_{N_2}(sz_N^{-1};sq^{-1}\beta^{-1})\left(|\Psi_{N_1}(\dots)\rangle \otimes K_{N_2}(s^{-1}z_N;sq^{-1}\beta^{-1})|\Psi_{N_2}(\dots,q^{-3}z_N)\rangle\right)\\
&= \langle\Psi_N(\dots,z_N^{-1})|K_N(sz_N^{-1};sq^{-1}\beta^{-1})K_N(s^{-1}z_N;sq^{-1}\beta^{-1})\left(|\Psi_{N_1}(\dots)\rangle \otimes |\Psi_{N_2}(\dots,q^{-3}z_N)\rangle\right).
\end{aligned}$$

The product of the $K$-matrices in the third line yields the identity matrix. Hence, we obtain

$$\Omega_{N_1,N_2}(\dots,z_N^{-1}) = \Omega_{N_1,N_2}(\dots,q^{-3}z_N),\tag{2.54}$$

which proves (2.49) for $i = N$. For $N_1 + 1 \leqslant i \leqslant N - 1$, the proof follows from Lemma 2.4. $\square$

In the next lemma, we relate the overlaps $\Omega_{N_1,N_2}$ and $\Omega_{N_2,N_1}$. The relation involves a transformation of the parameter $\beta$. Hence, to stress the dependence of the overlaps on this parameter, we write $\Omega_{N_1,N_2} = \Omega_{N_1,N_2}(z_1,\dots,z_{N_1},z_{N_1+1},\dots,z_N;\beta)$.

**Lemma 2.6.** *Let $s$ be a parameter with $s^2 = q^3$, then*

$$\begin{aligned}
\Omega_{N_1,N_2}(z_1,\dots,z_{N_1},z_{N_1+1},\dots,z_N;\beta)\\
= \Omega_{N_2,N_1}\left(sz_{N_1+1},\dots,sz_N,s^{-1}z_1,\dots,s^{-1}z_{N_1};sq^{-1}\beta^{-1}\right).\tag{2.55}
\end{aligned}$$

*Proof.* For each $N \geqslant 1$, we define the linear operator $\mathcal{P}$ on $V^N$ through

$$\mathcal{P}|\alpha_1\alpha_2\cdots\alpha_N\rangle = |\alpha_N\alpha_{N-1}\cdots\alpha_1\rangle,\tag{2.56}$$

for $\alpha_1,\dots,\alpha_N \in \{\uparrow,\downarrow\}$. For arbitrary $N \geqslant 1$, let us write $|\Psi_N\rangle = |\Psi_N(z_1,\dots,z_N;\beta)\rangle$ to stress the dependence on $\beta$. By [12, Proposition 3.15], we have

$$\mathcal{P}|\Psi_N(z_1,\dots,z_N;\beta)\rangle = |\Psi_N(s^{-1}z_N^{-1},\dots,s^{-1}z_1^{-1};sq^{-1}\beta^{-1})\rangle.\tag{2.57}$$

We apply this relation to the co-vector $\langle\Psi_N|$ in the definition of the overlap (2.41) and find

$$\begin{aligned}
\Omega_{N_1,N_2} &= \langle\Psi_N(s^{-1}q^3z_N,\dots,s^{-1}q^3z_{N_1+1},s^{-1}z_{N_1},\dots,s^{-1}z_1;sq^{-1}\beta^{-1})|\\
&\quad \times \mathcal{P}\left(|\Psi_{N_1}(z_1,\dots,z_{N_1};\beta)\rangle \otimes |\Psi_{N_2}(z_{N_1+1},\dots,z_N;\beta)\rangle\right).\tag{2.58}
\end{aligned}$$

Next, we simplify the second line of this equality with the help of the identity $\mathcal{P}(|\Phi_1\rangle \otimes |\Phi_2\rangle) = \mathcal{P}|\Phi_2\rangle \otimes \mathcal{P}|\Phi_1\rangle$. Applying (2.57), we obtain

$$\begin{aligned}
\Omega_{N_1,N_2} &= \langle\Psi_N(sz_N,\dots,sz_{N_1+1},q^{-3}sz_{N_1},\dots,q^{-3}sz_1;sq^{-1}\beta^{-1})|\\
&\quad \times \left(|\Psi_{N_2}(s^{-1}z_N^{-1},\dots,s^{-1}z_{N_1+1}^{-1};sq^{-1}\beta^{-1})\rangle \otimes |\Psi_{N_1}(s^{-1}z_{N_1}^{-1},\dots,s^{-1}z_1^{-1};sq^{-1}\beta^{-1})\rangle\right),\tag{2.59}
\end{aligned}$$

where we used $s^2 = q^3$. We compare the right-hand side with (2.41) and conclude

$$\begin{aligned}
\Omega_{N_1,N_2}(z_1,\dots,z_{N_1},z_{N_1+1},\dots,z_N;\beta)\\
= \Omega_{N_2,N_1}(s^{-1}z_N^{-1},\dots,s^{-1}z_{N_1+1}^{-1},s^{-1}z_{N_1}^{-1},\dots,s^{-1}z_1^{-1};sq^{-1}\beta^{-1}).\tag{2.60}
\end{aligned}$$

Finally, we use Lemmas 2.4 and 2.5, and once more $s^2 = q^3$, to obtain (2.55). $\square$

**Degree width**

The *degree width* of a Laurent polynomial is the difference in degree of its leading and trailing terms. Moreover, we call a Laurent polynomial *centred* if its degree width is twice the degree of its leading term. For instance, $z^3 - 2z^{-3}$ has degree width 6 and is centred. Conversely, $z^3 - 2z^{-2}$ has degree width 5 but is not centred.

**Lemma 2.7.** *For $N_1 \geq 1$ and each $1 \leq i \leq N_1$, $\Omega_{N_1,N_2}$ is a centred Laurent polynomial in $z_i$. Its degree width is at most $2(2N_1 + N_2 - 2)$.*

*Proof.* For each $N \geq 1$ and $1 \leq i \leq N$, each component of the vector $|\Psi_N\rangle$ is a Laurent polynomial in $z_i$. Hence, $\Omega_{N_1,N_2}$ is a Laurent polynomial in $z_i$ for each $1 \leq i \leq N_1$. By Lemma 2.5, it is centred. The Propositions 3.11, 3.12, and 3.14 of [12] imply that its degree width is at most $2(2N_1 + N_2 - 2)$. $\qquad\square$

For completeness, we add two observations concerning this lemma (which, however, will not be relevant in the following). First, we note that for odd $N_2$, the upper bound for the degree width can be sharpened to $2(2N_1 + N_2 - 3)$. Indeed, otherwise the overlap could have a leading term with an odd degree, which is incompatible with Lemma 2.3. Second, a similar result for $\Omega_{N_1,N_2}$ as a function of $z_i$ with $N_1 + 1 \leq i \leq N$ immediately follows from the combination of Lemmas 2.6 and 2.7.

**Reduction relations**

We now use the reduction relations for the vector $|\Psi_N\rangle$, given in (2.21) and Proposition 2.2, to derive reduction relations for the overlap $\Omega_{N_1,N_2}$. These relations are crucial to finding an explicit expression for the overlap at the combinatorial point.

**Lemma 2.8.** *For $N_1 \geq 2$ and each $2 \leq i \leq N_1$, we have*

$$\Omega_{N_1,N_2}(z_1 = q^{-1}z_i, \dots, z_i, \dots) = (-1)^{n+n_1}[q^2][\beta z_i][\beta q/z_i]/[q] \prod_{j=2, j\neq i}^{N_1}[qz_i/z_j][q^2/(z_iz_j)]$$

$$\times \prod_{j=2, j\neq i}^{N}[q^2z_j/z_i][qz_iz_j]\,\Omega_{N_1-2,N_2}(\widehat{z_1}, \dots, \widehat{z_i}, \dots), \quad (2.61)$$

*where $\widehat{z_j}$ denotes the omission of $z_j$, $n = \lfloor N/2 \rfloor$ and $n_1 = \lfloor N_1/2 \rfloor$.*

*Proof.* First, we prove the reduction relation for $i = 2$. Let us abbreviate

$$\Omega'_{N_1,N_2} = \Omega_{N_1,N_2}(q^{-1}z_2, z_2, \dots) = \Omega_{N_1,N_2}(z_2, q^{-1}z_2, \dots). \quad (2.62)$$

We find

$$\Omega'_{N_1,N_2} = \langle\Psi_N(z_2^{-1}, qz_2^{-1}, \dots)| \left(|\Psi_{N_1}(z_2, q^{-1}z_2, \dots)\rangle \otimes |\Psi_{N_2}(\dots)\rangle\right) \quad (2.63)$$

$$= (-1)^{n_1}[\beta z_2]\prod_{j=3}^{N_1}[q^2z_j/z_2][qz_2z_j]\langle\Psi_N(z_2^{-1}, qz_2^{-1}, \dots)|\Xi_N^1\left(|\Psi_{N_1-2}(\dots)\rangle \otimes |\Psi_{N_2}(\dots)\rangle\right).$$

From the first to the second line, we applied the reduction relation for $|\Psi_{N_1}(z_2, q^{-1}z_2, \dots)\rangle$. Next, we use $\Xi_N^1 = ([q^2]/(2[q]))\check{R}_{1,2}(q^{-1})\Xi_N^1$, the symmetry of the $\check{R}$-matrix and the exchange relations (2.14) to write

$$\langle\Psi_N(z_2^{-1}, qz_2^{-1}, \dots)|\Xi_N^1 = [q^2]/(2[q])\langle\Psi_N(qz_2^{-1}, z_2^{-1}, \dots)|\Xi_N^1. \quad (2.64)$$

We utilise the reduction relation of $\langle \Psi_N(qz_2^{-1}, z_2^{-1}, \dots)|$ on the right-hand side of this equality. It leads to

$$\langle \Psi_N(z_2^{-1}, qz_2^{-1}, \dots)|\Xi_N^1 = (-1)^n[q^2][\beta q/z_2]/[q]\prod_{j=3}^{N_1}[qz_2/z_j][q^2/(z_2z_j)]$$

$$\times \prod_{j=N_1+1}^{N}[q^2z_j/z_2][qz_2/z_j]\langle \Psi_{N-2}(\dots)|. \quad (2.65)$$

We combine this result with (2.63) and find

$$\Omega'_{N_1,N_2} = (-1)^{n+n_1}[q^2][\beta z_2][\beta q/z_2]/[q]\prod_{j=3}^{N_1}[qz_2/z_j][q^2/(z_2z_j)]$$

$$\times \prod_{j=3}^{N}[q^2z_j/z_2][qz_2/z_j]\Omega_{N_1-2,N_2}(\widehat{z_1},\widehat{z_2},\dots). \quad (2.66)$$

This result concludes the proof of (2.61) for $i = 2$. For $3 \leqslant i \leqslant N_1$, it follows from Lemma 2.4.

$\square$

## 2.4 The combinatorial point

In this section, and throughout the remainder of this article, we focus on the case where the crossing parameter takes the value

$$q = e^{2\pi i/3}. \quad (2.67)$$

We refer to this value as the *combinatorial point*. At this point, the vector $|\Psi_N\rangle$ is an eigenvector of the transfer matrix of a six-vertex model on a strip. Several properties of this eigenvector display relations to combinatorial problems [12].

The main result of this section is an explicit formula for the overlap $\Omega_{N_1,N_2}$ involving this eigenvector in terms of symplectic characters. We prove this formula through a standard strategy based on (strong) induction [37].

**Symplectic characters**

Let $N \geqslant 1$ and $\lambda_i = \lfloor (N-i)/2 \rfloor$ for each $1 \leqslant i \leqslant N$. We define the function $\chi_N$ through

$$\chi_N(z_1, \dots, z_N) = \frac{\det_{i,j=1}^N\left(z_j^{\mu_i} - z_j^{-\mu_i}\right)}{\det_{i,j=1}^N\left(z_j^{\delta_i} - z_j^{-\delta_i}\right)}, \quad (2.68)$$

where $\delta_i = N - i + 1$ and $\mu_i = \lambda_i + \delta_i$. This function is the symplectic character associated to the so-called *double-staircase partition* [38]. We simply refer to it as 'the symplectic character'. For convenience, we also define $\chi_0 = 1$.

The symplectic character $\chi_N$ is a symmetric function of $z_1, \dots, z_N$. With respect to each $z_i$, it is a centred Laurent polynomial of degree width $2(\bar{n}-1)$, where $\bar{n}$ is defined in (2.13), that is invariant under the reversal $z_i \to z_i^{-1}$. The coefficient of the leading term is itself a symplectic character. Indeed, the definition (2.68) implies that

$$\lim_{z_i \to \infty} z_i^{-(\bar{n}-1)}\chi_N(\dots, z_i, \dots) = \chi_{N-1}(\dots, \widehat{z_i}, \dots), \quad (2.69)$$

for each $N \geqslant 1$. A remarkable property of the symplectic character is the following reduction relation (see, for example, [39]): For $N \geqslant 2$ and $1 \leqslant i < j \leqslant N$, we have

$$\chi_N(\dots, z_i, \dots, z_j = q z_i, \dots) = \prod_{\substack{k=1 \\ k \neq i,j}}^{N} z_k^{-1}(z_k - q^2 z_i)(z_k - q z_i^{-1}) \chi_{N-2}(\dots, \widehat{z_i}, \dots, \widehat{z_j}, \dots). \quad (2.70)$$

**The overlap**

We now express the overlap $\Omega_{N_1, N_2}$ in terms of the symplectic character. To this end, we introduce the function $\bar{\Omega}_{N_1, N_2}$, defined through

$$\bar{\Omega}_{N_1, N_2}(z_1, \dots, z_{N_1}; z_{N_1+1}, \dots, z_N) = \epsilon_{N_1, N_2} \, \chi_{N_1}(z_1^2, \dots, z_{N_1}^2) \chi_{N_2}(z_{N_1+1}^2, \dots, z_N^2)$$
$$\times \chi_{N+1}(z_1^2, \dots, z_N^2, (\beta/q)^2), \quad (2.71)$$

where $N = N_1 + N_2$ and

$$\epsilon_{N_1, N_2} = \begin{cases} 0, & \text{for odd } N_1, N_2, \\ (-1)^{n+n_1 n_2}, & \text{otherwise}. \end{cases} \quad (2.72)$$

Here, $n$ is defined in (2.13) and

$$n_1 = \lfloor N_1/2 \rfloor, \quad n_2 = \lfloor N_2/2 \rfloor. \quad (2.73)$$

It is straightforward to check that $\bar{\Omega}_{N_1, N_2}$ has all the properties of $\Omega_{N_1, N_2}$ given in Lemmas 2.3 to 2.8, when specialised to $q = e^{2\pi i/3}$. Crucially, the reduction relations of Lemma 2.8 hold for $\bar{\Omega}_{N_1, N_2}$, too, thanks to the reduction relation (2.70) for the symplectic character.

**Theorem 2.9.** *For all $N_1, N_2 \geqslant 0$, we have $\Omega_{N_1, N_2} = \bar{\Omega}_{N_1, N_2}$.*

*Proof.* We note that the assertion of the theorem is equivalent to the statement that for all $M \geqslant 1$, we have $\Omega_{N_1, N_2} = \bar{\Omega}_{N_1, N_2}$ for all $N_1, N_2$ such that $N_1 + N_2 \leqslant M - 1$. We prove this statement by (strong) induction in $M$.

The base case of the induction is $M = 5$. One explicitly computes the vectors $|\Psi_1\rangle, \dots, |\Psi_4\rangle$ with the help of the contour-integral formulas of [12], and checks that $\Omega_{N_1, N_2} = \bar{\Omega}_{N_1, N_2}$, for all $N_1, N_2$ with $N_1 + N_2 \leqslant 4$. (For $N_1 + N_2 \leqslant 3$, this computation can even be done by hand with the components given at the end of Section 2.1.) Hence, we make the induction hypothesis that the statement hold for some integer $M = \bar{M} \geqslant 5$. Based on this assumption, we prove that $\Omega_{N_1, N_2} = \bar{\Omega}_{N_1, N_2}$ for all $N_1, N_2$ such that $N_1 + N_2 = \bar{M}$. To this end, we make the observation that at least one of the following cases always holds for $N_1 + N_2 = \bar{M} \geqslant 5$:

$$(i) \quad 6N_1 \geqslant 2\bar{M} + 5, \quad (ii) \quad 6N_2 \geqslant 2\bar{M} + 5. \quad (2.74)$$

We consider the two cases separately. Note that $N_1 > 2$ in case *(i)* and $N_2 > 2$ in case *(ii)*.

*Case (i)*: If $N_1, N_2$ are both odd then we trivially have $\Omega_{N_1, N_2} = \bar{\Omega}_{N_1, N_2} = 0$. Hence, let us suppose that at most one of the integers $N_1, N_2$ is odd. Combining Lemma 2.8 with Lemmas 2.3 to 2.5, we find that $\Omega_{N_1, N_2}$ can be reduced to $\Omega_{N_1-2, N_2}$ at the distinct points

$$z_1 = \pm q z_i, \pm q^{-1} z_i, \pm q z_i^{-1}, \pm q^{-1} z_i^{-1}, \quad 2 \leqslant i \leqslant N_1. \quad (2.75)$$

The same holds for $\bar{\Omega}_{N_1, N_2}$, too. By the induction hypothesis, we have $\Omega_{N_1-2, N_2} = \bar{\Omega}_{N_1-2, N_2}$. We conclude that $\Omega_{N_1, N_2} = \bar{\Omega}_{N_1, N_2}$ at the points (2.75). By Lemma 2.7, $\Omega_{N_1, N_2}$ is a centred Laurent polynomials in $z_1$ of degree at most $2(N_1 + \bar{M} - 2)$. The same holds for $\bar{\Omega}_{N_1, N_2}$ by virtue of the

properties of the symplectic character. Hence, these two Laurent polynomials are equal if the number of points (2.75) is greater or equal than (the upper bound for) the degree width plus one:

$$8(N_1 - 1) \geqslant 2(N_1 + \bar{M} - 2) + 1 \,. \tag{2.76}$$

This inequality indeed holds by virtue of the first inequality of (2.74). Hence, we have proven $\Omega_{N_1,N_2} = \bar{\Omega}_{N_1,N_2}$ for the case *(i)*.

*Case (ii)*: In this case, we use Lemma 2.6 with $s = 1$ to obtain

$$\Omega_{N_1,N_2}(z_1, \ldots, z_{N_1}; z_{N_1+1}, \ldots, z_N; \beta) = \Omega_{N_2,N_1}(z_{N_1+1}, \ldots, z_N; z_1, \ldots, z_{N_1}; q^{-1}\beta^{-1}) \,. \tag{2.77}$$

The same equality holds for $\bar{\Omega}_{N_1,N_2}$ by virtue of the properties of the symplectic character. We now consider both $\Omega_{N_1,N_2}$ and $\bar{\Omega}_{N_1,N_2}$ as functions of $z_{N_1+1}$ and repeat the argument of case *(i)*. This proves $\Omega_{N_1,N_2} = \bar{\Omega}_{N_1,N_2}$ for case *(ii)*, too.

Establishing the equality $\Omega_{N_1,N_2} = \bar{\Omega}_{N_1,N_2}$ for the two cases completes the induction step and concludes the proof of the theorem. $\qquad\square$

# 3 The homogeneous limit

In this section, we consider the so-called homogeneous limit where $z_1 = \cdots = z_N = 1$. The evaluation of $\Omega_{N_1,N_2}$ in this limit can be written in terms of the specialised symplectic character $\chi_N(1, \ldots, 1, z)$. In Section 3.1, we find a simple determinant formula for this specialised character. In Section 3.2, we obtain the spin-chain overlap $\mathcal{O}_{N_1,N_2}$ from the homogeneous limit of $\Omega_{N_1,N_2}$. Using the determinant formula for the specialised character, we prove Theorem 1.1.

## 3.1 Symplectic characters

**Multiple contour-integral formulas**

Using [40, Theorem 3.3], we write the symplectic character for $N = 2n + 1$ as

$$\chi_{2n+1}(z_1, \ldots, z_{2n}, z) = \frac{\prod_{i=1}^{2n} z_i^{-(n-1)} \prod_{i,j=1}^{n} h(z_i, z_{j+n})}{\bar{\Delta}(z_1, \ldots, z_n) \bar{\Delta}(z_{n+1}, \ldots, z_{2n})} \det_{i,j=1}^{n} \left( \frac{\chi_3(z_i, z_{j+n}, z)}{h(z_i, z_{j+n})} \right) \,. \tag{3.1}$$

Here, $\chi_3(z_1, z_2, z_3) = z_1 + z_1^{-1} + z_2 + z_2^{-1} + z_3 + z_3^{-1}$, $h(z, w) = (z^2 + zw + w^2)(1 + zw + z^2 w^2)$, and

$$\bar{\Delta}(z_1, \ldots, z_n) = \prod_{1 \leqslant i < j \leqslant n} (z_j - z_i)(1 - z_i z_j) \,. \tag{3.2}$$

The expression (3.1) is singular at $z_1 = \cdots = z_{2n} = 1$. To compute it at this point, we first rewrite the specialised character in terms of a multiple contour integral. For this purpose, we introduce the abbreviations

$$\mu(u, z) = (q^2 - z)u + q(1 - z), \quad f(u, z) = 3u - (1 - u + u^2)(1 + z + z^{-1}) \,, \tag{3.3}$$

where $q = e^{2\pi i/3}$, and use the Vandermonde

$$\Delta(z_1, \ldots, z_n) = \prod_{1 \leqslant i < j \leqslant n} (z_j - z_i) \,. \tag{3.4}$$

**Proposition 3.1.** *For each $n \geqslant 1$, we have the multiple contour-integral formula*

$$\chi_{2n+1}(z_1,\ldots,z_{2n},z) = \frac{3^{n(n-1)}\prod_{i,j=1}^n h(z_i,z_{j+n})}{n!\prod_{i=1}^{2n} z_i^{2n}} \oint_{C_1}\frac{du_1}{2\pi i}\cdots\oint_{C_n}\frac{du_n}{2\pi i}\prod_{i=1}^n f(u_i,z)u_i^{2(n-1)}$$

$$\times \frac{\Delta(u_1(u_1-1),\ldots,u_n(u_n-1))\Delta(u_1^{-1}(u_1^{-1}-1),\ldots,u_n^{-1}(u_n^{-1}-1))}{\prod_{i,j=1}^n \mu(u_i,z_j)\mu(u_i,z_j^{-1})\mu(u_i,q^2 z_{j+n})\mu(u_i,q^2 z_{j+n}^{-1})}, \quad (3.5)$$

*where the integration contour $C_i$ surrounds the poles $u_i = q(z_i-1)/(q^2-z_i)$ and $u_i = -q(z_i-1)/(q^2 z_i-1)$, but no other poles of the integrand.*

*Proof.* The proof is based on the following integral identity, similar to the identities used by de Gier, Pyatov and Zinn-Justin in [41]:

$$\frac{\chi_3(w,w',z)}{h(w,w')} = \frac{1}{w^2(w')^2}\oint_C \frac{du}{2\pi i}\frac{f(u,z)}{\mu(u,w)\mu(u,w^{-1})\mu(u,q^2 w')\mu(u,q^2(w')^{-1})}. \quad (3.6)$$

The integration contour $C$ on the right-hand side is a simple curve around the poles $u = q(w-1)/(q^2-w)$ and $u = -q(w-1)/(q^2 w-1)$, but no other poles of the integrand. The equality of both sides of (3.6) straightforwardly follows from the residue theorem.

We rewrite (3.1) with the help of (3.6) and obtain

$$\chi_{2n+1}(z_1,\ldots,z_{2n},z) = \frac{\prod_{i,j=1}^n h(z_i,z_{j+n})}{\prod_{i=1}^{2n} z_i^{n+1}\bar{\Delta}(z_1,\ldots,z_n)\bar{\Delta}(z_{n+1},\ldots,z_{2n})}$$

$$\times \det_{i,j=1}^n \left(\oint_{C_i}\frac{du}{2\pi i}\frac{f(u,z)}{\mu(u,z_i)\mu(u,z_i^{-1})\mu(u,q^2 z_{j+n})\mu(u,q^2 z_{j+n}^{-1})}\right). \quad (3.7)$$

Next, using Andreev's formula [42], we pull the contour integrals out of the determinant. This results in the multiple contour integral

$$\det_{i,j=1}^n \left(\oint_{C_i}\frac{du}{2\pi i}\frac{f(u,z)}{\mu(u,z_i)\mu(u,z_i^{-1})\mu(u,q^2 z_{j+n})\mu(u,q^2 z_{j+n}^{-1})}\right)$$

$$= \frac{1}{n!}\oint_{C_1}\frac{du_1}{2\pi i}\cdots\oint_{C_n}\frac{du_n}{2\pi i}\prod_{i=1}^n f(u_i,z)D(u_1,\ldots,u_n)\bar{D}(u_1,\ldots,u_n). \quad (3.8)$$

The right-hand side of this equality contains the two determinants

$$D(u_1,\ldots,u_n) = \det_{i,j=1}^n \left(\frac{1}{\mu(u_i,z_j)\mu(u_i,z_j^{-1})}\right), \quad (3.9)$$

$$\bar{D}(u_1,\ldots,u_n) = \det_{i,j=1}^n \left(\frac{1}{\mu(u_i,q^2 z_{j+n})\mu(u_i,q^2 z_{j+n}^{-1})}\right). \quad (3.10)$$

It was shown in [41] that they can be transformed into Cauchy determinants and, thus, be computed explicitly. The computation yields

$$D(u_1,\ldots,u_n) = \frac{(3q)^{n(n-1)/2}\prod_{i=1}^n u_i^{2(n-1)}\bar{\Delta}(z_1,\ldots,z_n)\Delta(u_1^{-1}(u_1^{-1}-1),\ldots,u_n^{-1}(u_n^{-1}-1))}{\prod_{i=1}^n z_i^{n-1}\prod_{i,j=1}^n \mu(u_i,z_j)\mu(u_i,z_j^{-1})},$$

$$\bar{D}(u_1,\ldots,u_n) = \frac{(3q^2)^{n(n-1)/2}\bar{\Delta}(z_{n+1},\ldots,z_{2n})\Delta(u_1(u_1-1),\ldots,u_n(u_n-1))}{\prod_{i=1}^n z_{i+n}^{n-1}\prod_{i,j=1}^n \mu(u_i,q^2 z_{j+n})\mu(u_i,q^2 z_{j+n}^{-1})}. \quad (3.11)$$

We evaluate (3.8) with the help of these expressions. Inserting the result into (3.7) leads to a cancellation of various pre-factors and yields (3.5). $\square$

To find the multiple contour-integral formula of the symplectic character with $N = 2n$, we combine the relation (2.69) and Proposition 3.1. A straightforward calculation leads to the following proposition:

**Proposition 3.2.** *For each $n \geq 1$, we have*

$$
\chi_{2n}(z_1, \ldots, z_{2n-1}, z) = \frac{3^{n(n-1)} \prod_{i,j=1}^{n} h(z_i, z_{j+n})}{(-q)^n n! \prod_{i=1}^{2n} z_i^{2n-1}} \oint_{C_1} \frac{du_1}{2\pi i} \cdots \oint_{C_n} \frac{du_n}{2\pi i} \prod_{i=1}^{n} \frac{f(u_i, z) u_i^{2(n-1)}}{1 - u_i + u_i^2}
$$

$$
\times \frac{\Delta(u_1(u_1-1), \ldots, u_n(u_n-1)) \Delta(u_1^{-1}(u_1^{-1}-1), \ldots, u_n^{-1}(u_n^{-1}-1))}{\prod_{i,j=1}^{n} \mu(u_i, z_j) \mu(u_i, z_j^{-1}) \prod_{i=1}^{n} \prod_{j=1}^{n-1} \mu(u_i, q^2 z_{j+n}) \mu(u_i, q^2 z_{j+n}^{-1})}, \quad (3.12)
$$

*where the integration contour $C_i$ surrounds the poles $u_i = q(z_i - 1)/(q^2 - z_i)$ and $u_i = -q(z_i - 1)/(q^2 z_i - 1)$, but no other poles of the integrand.*

### Determinant formulas

We now use Propositions 3.1 and 3.2 to obtain determinant formulas for $\chi_{2n+1}(1, \ldots, 1, z)$ and $\chi_{2n}(1, \ldots, 1, z)$. In view of our discussion of the homogeneous limit of $\Omega_{N_1, N_2}$, we consider $z = (\beta/q)^2$, and write the determinant formulas in terms of

$$
x = -[\beta q]/[\beta]. \tag{3.13}
$$

**Proposition 3.3.** *For $n \geq 0$, we have*

$$
\chi_{2n+1}(1, \ldots, 1, (\beta/q)^2) = \frac{3^{n^2}}{(1 - x + x^2)^n} \det_{i,j=1}^{n} \left( (x-1)^2 \binom{i+j-2}{2j-i-1} + x \binom{i+j}{2j-i} \right), \tag{3.14}
$$

*where $x$ is defined in (3.13).*

*Proof.* We have $\chi_1(z) = 1$ by straightforward computation, which proves the formula for $n = 0$. Hence, we focus on $n \geq 1$ and set $z_1 = \cdots = z_{2n} = 1$ in (3.5). The multiple contour integral simplifies to

$$
\chi_{2n+1}(1, \ldots, 1, z) = \frac{3^{n(n-1)}}{n!} \oint_0 \frac{du_1}{2\pi i} \cdots \oint_0 \frac{du_n}{2\pi i} \prod_{i=1}^{n} \frac{f(u_i, z)}{u_i^2} \Delta(u_1(u_1-1), \ldots, u_n(u_n-1))
$$

$$
\times \Delta(u_1^{-1}(u_1^{-1}-1), \ldots, u_n^{-1}(u_n^{-1}-1)). \tag{3.15}
$$

Here, the integration contour of each integral is a simple curve that surrounds the origin. We note that the integrand contains a product of Vandermonde determinants and, therefore, allows us to apply Andreev's formula [42]. Using this formula, we write the specialised symplectic character in terms of a single determinant:

$$
\chi_{2n+1}(1, \ldots, 1, z) = (-1)^{n(n-1)/2} 3^{n(n-1)} \det_{i,j=1}^{n} \left( \oint_0 \frac{du}{2\pi i} \frac{(u-1)^{i+j-2} f(u, z)}{u^{2j-i+1}} \right). \tag{3.16}
$$

Next, we set $z = (\beta/q)^2$. To write the specialised character in terms of $x$, we note that

$$
f(u, (\beta/q)^2) = \frac{3}{1 - x + x^2} \left( (x-1)^2 u - x(u-1)^2 \right). \tag{3.17}
$$

With the help of this equality, we obtain

$$
\chi_{2n+1}(1, \ldots, 1, (\beta/q)^2)
$$

$$
= \frac{(-1)^{n(n-1)/2} 3^{n^2}}{(1 - x + x^2)^n} \det_{i,j=1}^{n} \left( \oint_0 \frac{du}{2\pi i} \left( \frac{(u-1)^{i+j-2}}{u^{2j-i}} (x-1)^2 - \frac{(u-1)^{i+j}}{u^{2j-i+1}} x \right) \right). \tag{3.18}
$$

Finally, we evaluate the remaining contour integral inside the determinant with the help of

$$\oint_0 \frac{du}{2\pi i} \frac{(u-1)^k}{u^{\ell+1}} = (-1)^{k+\ell} \binom{k}{\ell}, \tag{3.19}$$

where $k, \ell$ are integers, and obtain (3.14). $\qquad\square$

**Proposition 3.4.** *For $n \geqslant 1$, we have*

$$\chi_{2n}(1,\ldots,1,(\beta/q)^2) = \frac{3^{n(n-1)}}{(1-x+x^2)^{n-1}} \det_{i,j=1}^{n-1} \left((x-1)^2\binom{i+j-1}{2j-i} + x\binom{i+j+1}{2j-i+1}\right), \tag{3.20}$$

*where $x$ is given by (3.13).*

*Proof.* The proof is similar to the proof of Proposition 3.3. First, we evaluate the integral (3.12) for $z_1 = \cdots = z_{2n-1} = 1$, which yields

$$\chi_{2n}(1,\ldots,1,z) = (-1)^{n(n-1)/2} 3^{n(n-2)} \det_{i,j=1}^{n} \left(\oint_0 \frac{du}{2\pi i} \frac{f(u,z)(u-1)^{i+j-2}}{u^{2j-i+1}(1-u+u^2)}\right). \tag{3.21}$$

To compute the remaining contour integral, we first note that

$$\frac{(u-1)^{i+j-2}}{u^{2j-i+1}(1-u+u^2)} = -\sum_{\ell=1}^{j} \frac{(u-1)^{i+\ell-3}}{u^{2\ell-i+1}} + \frac{u^{i-1}(u-1)^{i-2}}{1-u+u^2}. \tag{3.22}$$

The second term on the right-hand side is analytic at the point $u = 0$ for each $i \geqslant 1$. Hence, it does not contribute to the contour integral in (3.21). Thus, we may write

$$\chi_{2n}(1,\ldots,1,z) = (-1)^{n(n-1)/2} 3^{n(n-2)} \det_{i,j=1}^{n} \left(-\sum_{\ell=1}^{j} \oint_0 \frac{du}{2\pi i} \frac{f(u,z)(u-1)^{i+\ell-3}}{u^{2\ell-i+1}}\right) \tag{3.23}$$

$$= (-1)^{n(n+1)/2} 3^{n(n-2)} \det_{i,j=1}^{n} \left(\oint_0 \frac{du}{2\pi i} \frac{f(u,z)(u-1)^{i+j-3}}{u^{2j-i+1}}\right). \tag{3.24}$$

The evaluation of the integral inside the determinant with $z = (\beta/q)^2$ is similar to the evaluation at the end of the proof of Proposition 3.3. It leads to

$$\chi_{2n}(1,\ldots,1,(\beta/q)^2) = \frac{3^{n(n-1)}}{(1-x+x^2)^{n}} \det_{i,j=1}^{n} \left((x-1)^2\binom{i+j-3}{2j-i-1} + x\binom{i+j-1}{2j-i}\right). \tag{3.25}$$

Finally, we note that the entries of this determinant along the first row vanish unless $i = j = 1$. Expanding the determinant along this row leads to (3.20). $\qquad\square$

The specialisation of (3.14) and (3.20) to $\beta = q$ (corresponding to $x = 1$) leads to determinants of simple binomials. These determinants can be explicitly computed [43, Theorem 37 with $\mu = 0$ and $\mu = 1$]. The computation yields the following well-known result (see e.g. [40, 44]):

**Corollary 3.5.** *For each $N \geqslant 1$, we have $\chi_N(1,\ldots,1) = 3^{\nu_N}\gamma_N$, where $\gamma_N$ is defined in (1.9) and*

$$\nu_N = \begin{cases} n(n-1), & N = 2n, \\ n^2, & N = 2n+1. \end{cases} \tag{3.26}$$

## 3.2   The spin-chain overlap

The spin-chain ground state $|\psi_N\rangle$, defined in Section 1, is proportional to the homogeneous limit of the vector $|\Psi_N\rangle$. Indeed, using [12, (159)] specialised to $q = e^{2\pi i/3}$] we obtain

$$|\psi_N\rangle = (-1)^{n(n-1)/2} 3^{-\nu_N} [\beta]^{-n} |\Psi_N(1,\ldots,1;\beta)\rangle. \tag{3.27}$$

The left-hand side of this equality depends on the parameter $x$, whereas the right-hand side depends on $\beta$. The two parameters are related through (3.13).

*Proof of Theorem 1.1.*   First, we note that if at most one of the integers $N_1, N_2$ is odd, then the integers $n$ and $n_1, n_2$, defined in (2.13) and (2.73), respectively, obey $n = n_1 + n_2$. We use this relation and (3.27) to compute the overlap

$$\mathcal{O}_{N_1, N_2} = (-1)^{n_1 n_2} [\beta]^{-2n} 3^{-\nu_N - \nu_{N_1} - \nu_{N_2}} \Omega_{N_1, N_2}(1,\ldots,1;1,\ldots,1;\beta). \tag{3.28}$$

Next, we write the right-hand side of this expression in terms of symplectic characters with the help of Theorem 2.9. Moreover, we rewrite the pre-factor in terms of $x$, instead of $\beta$, and obtain

$$\mathcal{O}_{N_1, N_2} = 3^{-\nu_{N_1}} \chi_{N_1}(1,\ldots,1) 3^{-\nu_{N_2}} \chi_{N_2}(1,\ldots,1)$$
$$\times 3^{-\nu_{N+1}} (1 - x + x^2)^n \chi_{N+1}(1,\ldots,1,(\beta/q)^2). \tag{3.29}$$

Second, we evaluate the symplectic characters. The evaluation depends on the parities of the integers $N_1, N_2$. First, for even $N_1, N_2$, Proposition 3.3 and Corollary 3.5 allow us to write

$$\mathcal{O}_{N_1, N_2} = \gamma_{N_1} \gamma_{N_2} \det_{i,j=1}^{n} \left( (x-1)^2 \binom{i+j-2}{2j-i-1} + x \binom{i+j}{2j-i} \right). \tag{3.30}$$

Second, if $N_1$ is even and $N_2$ is odd, then we use Proposition 3.4 and Corollary 3.5 to obtain

$$\mathcal{O}_{N_1, N_2} = \gamma_{N_1} \gamma_{N_2} \det_{i,j=1}^{n} \left( (x-1)^2 \binom{i+j-1}{2j-i} + x \binom{i+j+1}{2j-i+1} \right). \tag{3.31}$$

Third, if $N_1$ is odd but $N_2$ even, we obtain same the expression.   □

The determinant expressions (3.30) and (3.31) for the spin-chain overlaps were conjectured in [12]. Using recent results by Fischer and Saikia [45], one can even compute the determinants explicitly in terms of integers enumerating alternating sign matrices and rhombus tilings. Indeed, using [45, (3.9) and (3.10)], we find

$$\det_{i,j=1}^{n} \left( (x-1)^2 \binom{i+j-1}{2j-i} + x \binom{i+j+1}{2j-i+1} \right) = \sum_{i=0}^{2n} A_O(2n+2, i+2) x^i, \tag{3.32}$$

where $A_O(2n, i)$ is the number of off-diagonally symmetric alternating sign matrices of size $2n \times 2n$, whose unique entry $+1$ in the first row is at position $i$. We have $A_O(2n, 1) = 0$ and, for $i \geqslant 2$, the expression

$$A_O(2n, i) = A_V(2n-1) \sum_{k=1}^{i-1} (-1)^{i+k-1} \frac{(2n+k-2)!(4n-k-1)!}{(4n-2)!(k-1)!(2n-k)!}. \tag{3.33}$$

Similarly, it is possible to compute the determinant (3.30), using [45, (4.7) and (4.8)]. The coefficients of the resulting polynomial are integers that can, in principle, be expressed in terms of numbers $Q_{n,i}$, which enumerate certain rhombus tilings. The explicit expressions are, however, quite involved, and we do not report them here.

# 4 The asymptotic series

The aim of this section is to compute the leading terms of the asymptotic series of $\mathcal{F}_{N_1,N_2}$ as $N_1, N_2 \to \infty$. We prepare this computation in Section 4.1 with a discussion of the asymptotic properties of the symplectic character, using results obtained by Gorin and Panova [30]. These asymptotic properties allow us to prove Theorem 1.2 in Section 4.2. In Section 4.3, we compare the leading terms of the asymptotic series for the LBF to the CFT predictions.

## 4.1 The symplectic character

Let us introduce the normalised symplectic character

$$\mathfrak{X}_N(z) = \frac{\chi_N(1,\ldots,1,z)}{\chi_N(1,\ldots,1)}. \tag{4.1}$$

Gorin and Panova showed that

$$\mathfrak{X}_N(z) = \frac{3z^{3/4}(z-1)}{(z^{3/2}-1)(z+1)} \left( \frac{4}{9} \frac{(z^{3/2}-1)^2}{z^{1/2}(z-1)^2} \right)^N$$

$$\times \begin{cases} 1 + \tau_1(z)N^{-1/2} + \tau_2(z)N^{-1} + o(N^{-1}), & \text{even } N, \\ 1 + \bar{\tau}_1(z)N^{-1/2} + \bar{\tau}_2(z)N^{-1} + o(N^{-1}), & \text{odd } N, \end{cases} \tag{4.2}$$

asymptotically as $N \to \infty$ [30, Proposition 5.13].[3]

We need the explicit expressions of the coefficients $\tau_i(z)$, $\bar{\tau}_i(z)$, $i = 1, 2$, in the sub-leading terms. Following the strategy of [29], we compute these coefficients with the help of a differential equation. We write this differential equation for the function $f_N$ defined through

$$f_N(z) = z^{-N}(z-1)^{2N-1}(z+1)\mathfrak{X}_N(z). \tag{4.3}$$

**Proposition 4.1.** *For each $n \geqslant 0$, we have*

$$z\frac{\mathrm{d}}{\mathrm{d}z}\left(zf'_{2n}(z)\right) + 3(2n-1)\frac{1+z^3}{1-z^3}zf'_{2n}(z) + (3n-1)(3n-2)f_{2n}(z) = 0, \tag{4.4}$$

$$z\frac{\mathrm{d}}{\mathrm{d}z}\left(zf'_{2n+1}(z)\right) + 6n\frac{1+z^3}{1-z^3}zf'_{2n+1}(z) + (3n+1)(3n-1)f_{2n+1}(z) = 0. \tag{4.5}$$

*Proof.* By [30, Theorem 3.18], we have the contour-integral representation

$$f_N(z) = 2(2N-1)! \oint_{C^+} \frac{\mathrm{d}w}{2\pi\mathrm{i}} \frac{z^w - z^{-w}}{\prod_{i=1}^N (w^2 - \mu_i^2)}. \tag{4.6}$$

Here, the integration contour $C^+$ is a simple curve that surrounds the poles at $w = \mu_i$, but not the poles at $w = -\mu_i$. (Recall that $\mu_i = \lambda_i + N - i + 1$, $\lambda_i = \lfloor (N-i)/2 \rfloor$.)

We now define a differential operator $A$ through

$$Ag(z) = z\frac{\mathrm{d}}{\mathrm{d}z}\left(zg'(z)\right) + (\mu_1 + \mu_2)\frac{1+z^3}{1-z^3}zg'(z) + \mu_1\mu_2 g(z). \tag{4.7}$$

Its action on the $z$-dependent part of the integrand of (4.6) yields

$$A(z^w - z^{-w}) = \frac{(w+\mu_1)(w+\mu_2)(z^{3-w}+z^w) - (w-\mu_1)(w-\mu_2)(z^{3+w}+z^{-w})}{1-z^3}. \tag{4.8}$$

---

[3] The expression given here corrects a typo in [30], where the first term is given with a power $z^{-9/4}$, instead of $z^{3/4}$. We note that the first line of (4.2) is invariant under $z \to z^{-1}$, as expected from the invariance of the symplectic character under this substitution.

Using this relation, we obtain the action of $A$ on $f_N$:

$$Af_N(z) = \frac{2(2N-1)!}{1-z^3} \oint_C \frac{dw}{2\pi i} F(z,w). \tag{4.9}$$

Here, the contour integral contains the function

$$F(z,w) = \frac{z^{3-w}+z^w}{(w-\mu_1)(w-\mu_2)\prod_{i=3}^N(w^2-\mu_i^2)}. \tag{4.10}$$

The integration contour $C$ is a simple curve around *all* the poles $\mu_1,\dots,\mu_N,-\mu_3,\dots,-\mu_N$ of the integrand.

Finally, we consider the reflection $\varphi(w) = 3-w$. It maps the set of poles of the integrand onto itself. Moreover, one checks that $F(z,\varphi(u)) = F(z,u)$. The change of variables $w = \varphi(u)$ in (4.9) leads to

$$Af_N(z) = -\frac{2(2N-1)!}{1-z^3}\oint_{\varphi(C)}\frac{du}{2\pi i}F(z,\varphi(u)) = -\frac{2(2N-1)!}{1-z^3}\oint_{\varphi(C)}\frac{du}{2\pi i}F(z,u). \tag{4.11}$$

The image integration contour $\varphi(C)$ is a simple curve that surrounds all the poles of the integrand. Hence, it can be deformed into $C$. This deformation results in $Af_N(z) = -Af_N(z)$. Hence, we obtain the differential equation

$$Af_N(z) = 0. \tag{4.12}$$

The proposition follows from writing out this differential equation with $\mu_1 = 3n-1, \mu_2 = 3n-2$ for $N = 2n$, and $\mu_1 = 3n+1$, $\mu_2 = 3n-1$ for $N = 2n+1$. $\qquad\square$

**Lemma 4.2.** *We have*

$$\tau_1(z) = 0, \quad \tau_2(z) = -\frac{5}{144}\frac{(z^{3/2}-1)^2}{z^{3/2}}, \tag{4.13}$$

$$\bar{\tau}_1(z) = 0, \quad \bar{\tau}_2(z) = +\frac{7}{144}\frac{(z^{3/2}-1)^2}{z^{3/2}}. \tag{4.14}$$

*Proof.* We substitute the asymptotic expansion (4.2) into the differential equations of Proposition 4.1. For $N = 2n$, we find

$$\sqrt{2}\tau_1'(z)n^{1/2} + \left(\tau_2'(z)-\tau_1'(z)\tau_1(z)+\frac{5}{96}\frac{z^3-1}{z^{5/2}}\right) + O(n^{-1/2}) = 0, \tag{4.15}$$

asymptotically as $n \to \infty$. Equating the coefficients of the two leading terms to zero, we find two differential equations for $\tau_1$ and $\tau_2$. Their solution is

$$\tau_1(z) = c_1, \quad \tau_2(z) = c_2 - \frac{5}{144}\frac{z^3+1}{z^{3/2}}, \tag{4.16}$$

where $c_1$ and $c_2$ are integration constants. Since $\mathfrak{X}_N(1) = 1$ for all $N$, we have $\tau_1(1) = \tau_2(1) = 0$. This initial condition fixes the values of the integration constants to $c_1 = 0$ and $c_2 = 5/72$ and leads to the expressions given in (4.13).

Similarly, the expressions (4.14) follow from the differential equation for $N = 2n+1$. $\quad\square$

## 4.2 The logarithmic bipartite fidelity

Let us write $\mathcal{F}_{N_1,N_2} = \mathcal{F}_{N_1,N_2}(x)$ to stress the dependence of the LBF on the parameter $x$. Using (1.6), (3.29) and (4.1), we find

$$\mathcal{F}_{N_1,N_2}(x) = \mathcal{F}_{N_1,N_2}(1) - \ln\left(\frac{\mathfrak{X}_{N+1}(z)}{\mathfrak{X}_{N_1+1}(z)\mathfrak{X}_{N_2+1}(z)}\right), \tag{4.17}$$

where $N = N_1 + N_2$ and

$$z = \left(\frac{\beta}{q}\right)^2 = \frac{qx+1}{q+x}. \tag{4.18}$$

*Proof of Theorem 1.2.* First, we determine the leading terms of the asymptotic series of $\mathcal{F}_{N_1,N_2}(1)$ as $N_1, N_2 \to \infty$. It was computed in [11] up to $O(1)$ with the help of Stirling's formula. Finding the $O(N^{-1})$-term is straightforward. If $N_1, N_2$ are even, then

$$\mathcal{F}_{N_1,N_2}(1) = \frac{1}{6}\ln N + \frac{1}{6}\ln(\xi(1-\xi)) - \ln K + \frac{13}{72}\left(\frac{1}{\xi} + \frac{1}{1-\xi} - 1\right)N^{-1} + O(N^{-2}), \tag{4.19}$$

where

$$K = \frac{8}{3\Gamma(1/3)}\sqrt{\frac{\pi}{3}}. \tag{4.20}$$

Likewise, for even $N_1$ and odd $N_2$, we obtain

$$\mathcal{F}_{N_1,N_2}(1) = \frac{1}{6}\ln N + \frac{1}{6}\ln\left(\frac{\xi}{1-\xi}\right) - \ln K + \frac{1}{72}\left(\frac{13}{\xi} + 11\left(1 - \frac{1}{1-\xi}\right)\right)N^{-1} + O(N^{-2}). \tag{4.21}$$

Second, we consider $z$ defined in (4.18). Using the parameterisation (1.13), we find $z = e^{2\pi i(1-r)/3}$. We evaluate the second term of (4.17) for this value of $z$ with the help of the asymptotic series (4.2). For even $N_1, N_2$, we find

$$\ln\left(\frac{\mathfrak{X}_{N+1}(z)}{\mathfrak{X}_{N_1+1}(z)\mathfrak{X}_{N_2+1}(z)}\right) = \ln\left(\frac{3\sin(2\pi(r-1)/3)}{4\sin(\pi(r-1)/2)}\right)$$

$$+ \bar{\tau}_2(z)\left(1 - \frac{1}{\xi} - \frac{1}{1-\xi}\right)N^{-1} + o(N^{-1}). \tag{4.22}$$

For even $N_1$ and odd $N_2$, we obtain

$$\ln\left(\frac{\mathfrak{X}_{N+1}(z)}{\mathfrak{X}_{N_1+1}(z)\mathfrak{X}_{N_2+1}(z)}\right) = \ln\left(\frac{3\sin(2\pi(r-1)/3)}{4\sin(\pi(r-1)/2)}\right)$$

$$+ \left(\tau_2(z)\left(1 - \frac{1}{1-\xi}\right) - \frac{\bar{\tau}_2(z)}{\xi}\right)N^{-1} + o(N^{-1}). \tag{4.23}$$

Using Lemma 4.2, we infer

$$\tau_2(z) = \frac{5}{36}\sin^2\left(\frac{\pi(r-1)}{2}\right), \quad \bar{\tau}_2(z) = -\frac{7}{36}\sin^2\left(\frac{\pi(r-1)}{2}\right). \tag{4.24}$$

Inserting (4.19), (4.20) and (4.22) into (4.17) leads to (1.16), whereas (4.21), (4.20) and (4.23) allow us to obtain (1.17).

Finally, for odd $N_1$ and even $N_2$, we exploit the fact that $\mathcal{F}_{N_1,N_2} = \mathcal{F}_{N_2,N_1}$ (which follows from (1.6) and Theorem 1.1). This property implies that we may obtain the asymptotic series from the one for even-odd case, with $\xi$ replaced by $1 - \xi$. □

### 4.3 The prediction of conformal field theory

We now compare Theorem 1.2 to the CFT prediction for the asymptotic series of the LBF. For the case (1.3), we expect this CFT to be a free boson theory, whose compactification radius is fine-tuned to a value where the conformal symmetry is enhanced to a superconformal symmetry [16, 46]. Here below, we recall the CFT prediction for this case and use physics arguments to infer the CFT data from the characteristics of the spin chain's ground-state vector.

**The CFT prediction**

Dubail and Stéphan computed the leading terms of the asymptotic series of $\mathcal{F}_{N_1,N_2}$ for one-dimensional quantum critical systems using CFT arguments [7, 9]. Their CFT derivation relates the LBF to a correlation function of four primary boundary fields $\varphi_i$ with conformal weights $\Delta_i$, $1 \leqslant i \leqslant 4$, on a so-called *flat-pants domain*, as illustrated in Figure 2. The prediction is

$$\mathcal{F}_{N_1,N_2} = \left(\frac{c}{8} + \Delta_2\right)\ln N + f(\xi) + g(\xi)N^{-1}\ln N + O(N^{-1}), \qquad (4.25)$$

asymptotically as $N_1, N_2 \to \infty$, where $N = N_1 + N_2$ and $\xi = N_1/N$. The coefficient of the leading term is universal. It only depends on the CFT's central charge $c$ and the conformal weight $\Delta_2$ of the primary field localised at tip of the slit in the flat-pants domain. The coefficients $f(\xi)$ and $g(\xi)$ of the sub-leading terms possess explicit expressions in terms of the fields' conformal weights. We present these expressions here below for a free boson theory with central charge $c = 1$. For this theory, the weights are $\Delta_i = \alpha_i^2/2$ where $\alpha_i$ are the fields' so-called $U(1)$ charges. We have

$$f(\xi) = \left(\frac{1}{24}\left(2\xi - 1 + \frac{2}{\xi}\right) + \left(1 - \frac{1}{\xi}\right)\alpha_1^2 - \frac{\alpha_2^2}{2} - 2\alpha_2\alpha_3 - \alpha_3^2 + (1-\xi)\alpha_4^2\right)\ln(1-\xi) \quad (4.26)$$

$$+ \left(\frac{1}{24}\left(1 - 2\xi + \frac{2}{1-\xi}\right) + \left(1 - \frac{1}{1-\xi}\right)\alpha_3^2 - \frac{\alpha_2^2}{2} - 2\alpha_2\alpha_1 - \alpha_1^2 + \xi\alpha_4^2\right)\ln\xi + C. \quad (4.27)$$

Here, $C$ is a non-universal constant in the sense that it cannot be obtained from CFT arguments. The coefficient $g(\xi)$ is given by

$$g(\xi) = \Xi \times \frac{1}{2}\left(\alpha_4^2 - \frac{1}{12} + \left(\frac{1}{12} - \alpha_1^2\right)\frac{1}{\xi} + \left(\frac{1}{12} - \alpha_3^2\right)\frac{1}{1-\xi}\right), \qquad (4.28)$$

where $\Xi$ is a non-universal factor called the *extrapolation length* [47, 48]. We note that the $U(1)$ charges, as well as the non-universal constants $C$ and $\Xi$ may in principle depend on the parity of the integers $N_1, N_2$.

**Comparison and discussion**

To compare Theorem 1.2 to the CFT prediction, we need to find the charges $\alpha_1, \ldots, \alpha_4$ of the four primary fields on the flat pants domain. To this end, we follow the analysis of [11] and consider the asymptotic series of the ground-state eigenvalue of the spin-chain Hamiltonian as $N \to \infty$. For a general quantum critical Hamiltonian with open boundary conditions, the first terms of this series are expected to be [49, 50]

$$E_0 = NE_{\text{bulk}} + E_{\text{bndr}} + \pi v_F\left(\Delta_0 - \frac{c}{24}\right)N^{-1} + O(N^{-2}). \qquad (4.29)$$

Here, the pre-factors $E_{\text{bulk}}$ and $E_{\text{bndr}}$ are non-universal numbers. Moreover, the so-called Fermi velocity $v_F$ is a non-zero number. For the XXZ chain, it can be written in terms of the anisotropy

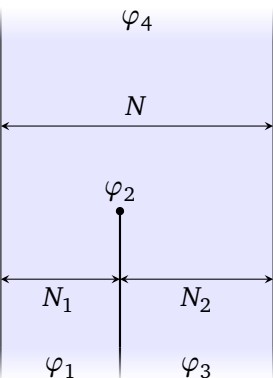

Figure 2: The picture illustrates the geometry of the flat-pants domain. It corresponds to a concatenation of an upper vertical semi-infinite strip of width $N$ and two lower vertical semi-infinite strips of widths $N_1, N_2$, separated by a slit. The LBF can be constructed from CFT correlation functions with primary fields at boundary points of this domain. The fields $\varphi_1$, $\varphi_3$ and $\varphi_4$ are localised at the infinitely-remote ends of the pants' legs, whereas the insertion point of $\varphi_2$ is the top of the slit.

parameter $\Delta$ [51]. Finally, $\Delta_0$ is the conformal weight of the primary field associated to the scaling limit of the spin chain's ground state. For the Hamiltonian of the open XXZ chain (1.1) with the parameters (1.3), we infer from the explicit expression (1.4) of $E_0$ the values

$$E_{\text{bulk}} = -\frac{3}{4}, \quad E_{\text{bndr}} = \frac{1}{4}(2-x)(2-x^{-1}), \quad \Delta_0 = \frac{1}{24}. \tag{4.30}$$

Here, we assumed that the central charge is $c = 1$. Since the primary fields $\varphi_1, \varphi_3, \varphi_4$ correspond to the ground state, we immediately infer the conformal weights $\Delta_1 = \Delta_3 = \Delta_4 = \frac{1}{24}$, independently of the value of $x > 0$. These values fix the corresponding $U(1)$ charges up to a sign. For $x = 1$, the analysis of the magnetisation of the ground-state vectors involved in the spin-chain overlap determines this sign [11]. Since the magnetisation of these vectors does not change with $x$ (and since the Hamiltonian depends continuously on $x$), it is plausible to assume that the unknown sign coincides with its value at $x = 1$. For even $N_1, N_2$, this assumption leads to[4]

$$\alpha_1 = \alpha_3 = \frac{1}{2\sqrt{3}}, \quad \alpha_4 = -\frac{1}{2\sqrt{3}}, \tag{4.31}$$

whereas, for even $N_1$ and odd $N_2$, we find the values

$$\alpha_1 = \alpha_4 = \frac{1}{2\sqrt{3}}, \quad \alpha_3 = -\frac{1}{2\sqrt{3}}. \tag{4.32}$$

The $U(1)$ charge of the primary field at the tip of the cut follows from the charge neutrality condition $\alpha_1 + \alpha_2 + \alpha_3 + \alpha_4 = 0$. In both cases, we obtain

$$\alpha_2 = -\frac{1}{2\sqrt{3}}. \tag{4.33}$$

Upon substitution of the $U(1)$ charges (4.31) and (4.33) into (4.26) and (4.28), we obtain for even $N_1, N_2$ the expressions

$$f(\xi) = \frac{1}{6} \ln(\xi(1-\xi)) + C_{\text{ee}}, \quad g(\xi) = 0. \tag{4.34}$$

---

[4]The sign convention for the $U(1)$ charges used here differs slightly from [11]. That article exclusively works with $U(1)$ charges assigned to vectors (kets). Here, in contrast, $\alpha_4$ is the $U(1)$ charge of a co-vector (bra), which has the advantage that the charge neutrality condition matches the conventions commonly used in CFT. Note that the charges of a ket and a bra vector are obtained from one another through the multiplication by a minus sign.

Similarly, using (4.32) and (4.33), we find

$$f(\xi) = \frac{1}{6} \ln\left(\frac{\xi}{1-\xi}\right) + C_{\text{eo}}, \quad g(\xi) = 0, \tag{4.35}$$

for even $N_1$ and odd $N_2$. Comparing these expressions with Theorem 1.2, we observe that the CFT prediction perfectly matches the exact calculations, provided that we set $C_{\text{ee}} = C_{\text{eo}} = -\ln D$. In particular, the vanishing of the coefficient $g(\xi)$ in both cases is consistent with the identification of the conformal weights and $U(1)$ charges associated to the ground states.

We would like to point out the power of the CFT results. Based on a few physical arguments, they effortlessly predict that the leading terms of the LBF's asymptotic series do not depend on the boundary parameter $x > 0$, except for a non-universal additive constant in the $O(1)$-term that may depend on it. This is by no means obvious from the finite-size expressions which have a highly non-trivial $x$-dependence.

## 5 Conclusion

In this article, we have computed the finite-size LBF for the open XXZ spin chain with anisotropy $\Delta = -\frac{1}{2}$ and a one-parameter family of diagonal boundary magnetic fields. The computation exploits the fact that the spin chain's ground-state vector can be obtained as a specialisation of a solution to boundary quantum Knizhnik-Zamolodchikov equations. We have used the finite-size results to find the asymptotic series of the LBF for large systems. The leading terms of this series perfectly match the CFT predictions.

We conclude our investigation with a short discussion of possible generalisations of the present work and open problems. First, the LBF of the XXZ chain can be generalised in several ways. These generalisations are based on bipartite and multipartite overlaps, involving the ground-state vectors for periodic, twisted and open boundary conditions. For the anisotropy parameter $\Delta = -\frac{1}{2}$, many of these generalisations possess exact finite-size expressions in terms of combinatorial numbers. They allow one to find the corresponding fidelities' asymptotic series for large systems [13, Chapter 7]. However, a CFT prediction for these asymptotic series, in particular for multipartite fidelities, appears to be missing in the literature and could be an interesting endeavour. Second, the techniques used to find bipartite and multipartite overlaps and fidelities for the XXZ chain at $\Delta = -\frac{1}{2}$, such as supersymmetry or the quantum Knizhnik-Zamolodchikov equations, do not easily generalise to other values of the anisotropy parameter $\Delta$. However, for generic values of this parameter, the standard techniques of quantum integrability, such as the algebraic Bethe ansatz or the quantum separation of variables technique allow one, in principle, to construct the spin chain's ground state. It remains a challenge to understand if these constructions lead to expressions for the LBF that are amenable to an asymptotic analysis and the extraction of the asymptotic series' leading coefficients. Third, very few properties of the bipartite fidelity for off-critical or near-critical systems are known. To leading order, the LBF of an off-critical system is characterised by a correlation length [7,8]. However, in contrast to critical systems, nothing appears to be known about sub-leading terms and their possible physical content. Moreover, it remains an open problem to understand if the LBF of a near-critical system can be characterised in terms of a massive quantum field theory. A suitable starting point to investigate these questions could be the computation of the LBF for the open XY spin chain. Alternatively, one might investigate the supersymmetry-preserving deformation of the supersymmetric point $\Delta = -1/2$, $p = \bar{p} = -1/4$ to the open XYZ chain [18]. As for the XXZ case, the exact lattice supersymmetry could simplify the evaluation of bipartite and multipartite overlaps and open up the possibility for an exact calculation of the LBF.

## Acknowledgements

This work was supported by the Fonds de la Recherche Scientifique (F.R.S.-FNRS) and the Fonds Wetenschappelijk Onderzoek-Vlaanderen (FWO) through the Belgian Excellence of Science (EOS) project no. 30889451 "PRIMA – Partners in Research on Integrable Models and Applications". CH was supported in part by the Centre de la Recherche Scientifique (CNRS) and thanks the Laboratoire de Physique Théorique et Modèles Statistiques, Université Paris-Saclay, for hospitality. GP holds a CRM-ISM postdoctoral fellowship and acknowledges support from the Mathematical Physics Laboratory of the CRM. We thank Alexi Morin-Duchesne, Jean Liénardy and Jean-Marie Stéphan for discussions, and Alexandre Lazarescu for his comments on the manuscript.

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
