# Peer review of "On the logarithmic bipartite fidelity of the open XXZ spin chain at $Δ=-1/2$"

_SciPost Physics, doi:SciPost Phys. 12, 199 (2022)_

## Round 1 · Referee Report · Anonymous · 2022-2-14

Strengths

1- Neat exact computations of a universal quantity.
2- Clearly written.

Weaknesses

1- The generalisation of previous results considered here does not affect the universal behavior, compared to previous studies.

Report

In this paper, the authors study the 'logarithmic bipartite fidelity' in the spin 1/2-XXZ spin chain. This is, roughly speaking, the overlap between the ground state of a quantum system and the tensor product of ground states of smaller systems. It can be studied using conformal field theory (CFT) methods, in the spirit of Fermi edge singularities and impurity problems often encountered in condensed matter theory. These techniques allow to make various asymptotic predictions for large systems, in particular the orthogonality catastrophe exponent, and sometimes the form of subleading corrections.

Here the focus is on exact (and even rigorous) lattice computations using quantum integrability techniques, and a precise asymptotic analysis which confirms CFT predictions. While performing this task is difficult in full generality, the authors manage to do so at the 'Razumov-Stroganov' point $\Delta=-1/2$ and for a simple family of boundary fields. The main result takes the form of two theorems, one providing an exact determinant formula for the overlaps which was previously conjectured by one of the authors, another for the asymptotic expansion using a relation to the theory of symplectic characters.

Overall the paper is clearly written and a pleasant read. The results are nontrivial, I recommend publication, provided the authors address the minor issues which are listed below.

Requested changes

1) Page 3, after equation (1.8), 'definite magnetisation' reads awkwardly.

2) Page 4, theorem 1.1. This is stated later, but it would be better to already mention that this was conjectured in reference [12].

3) In theorem 1.2, I think it is implied that $\xi$ is in $(0,1)$, as one could imagine other asymptotic regimes, e.g $N_2\to\infty$ first, and then $N_1\to \infty$.

4) In section 2.1, the authors make an effort to be very precise, but they might want to already comment on their convention not to take complex conjugate for the bra, which is non-standard in quantum mechanics. This would help later on for certain calculations and understanding footnote 4. For equation (2.10): even though this is well-known, add a sentence explaining what is meant by the subscripts, similar to say (2.4).

5) While reading through the introduction, it is not very clear at this stages where equation (2.15) does come from.

6) After equation (3.7). It very much looks like it is Andreev's formula.

7) Proposition 3.3. This is fine as it is, but wouldn't it be also possible to use the method of reference [35]? That is make row manipulations in the determinant, and use Taylor expansions to take the limit.

8) Around Lemma 4.2, can the authors comment on the fact that $\tau_1$ and $\bar{\tau}_1$ actually vanish? Is there some simple symmetry reason which might explain it? Two related questions: do we expect corrections of the form $N^{-3/2}, N^{-5/2},\ldots$ or should all these terms vanish? Is (4.4), (4.5) sufficient to imply $g(\xi)=0$ without relying on the results of reference [30]?

9) Before equation (4.31). Replace 'it is plausible assume' by 'it is plausible to assume'.

---

## Round 1 · Referee Report · Anonymous · 2022-4-12

Report

This paper is devoted to the exact computation of the finite size bipartite fidelity of the open XXZ spin chain at the anisotropy parameter ∆ = − 12 . The asymptotic behavior of exact rigorous result is compared to the (more general) CFT prediction and perfect agreement is found.

The finite size computation is made possible thanks to the fact that the ground state of the open XXZ chain (at ∆ = − 12 ) and boundary parameters defined in eq.(1.3) can be obtained
as the homogeneous limit of the (minimal degree?) solution of the boundary qKZ equation (at q = e^2iπ/3 ). This fact is mentioned only en passant at the beginning of section 2.4 and I believe it should be more emphasized. Maybe they should state explicitly something like
|Ψ N (z 1 , . . . , z N )> =1 ∝ |ψ N >.
Working with the solution of the bqKZ equation allows the authors to defined a spectral parameter deformation of the overlaps appearing in the computation of the bipartite fidelity. These deformed overlaps are completely characterized by the symmetries and the analytical dependence on the spectral parameters. This allows to show that they can be expressed in terms of certain symplectic characters. Only then, the homogeneous limit (z i = 1) is taken.

A question came to my mind. Is it maybe possible to obtain formula (3.14) using the
Jacobi–Trudi or the Giambelli identity for symplectic characters 1 ?

In conclusion: the paper is carefully written and pleasant to read. It presents new and in my opinion interesting results, that as far as I’ve been able to check are correct. I recommend the paper for publication.

---

## Round 1 · Referee Report · Anonymous · 2022-4-22

Report

It is known, based on initial observations of Razumov and Stroganov from about 20 years ago and much subsequent work, that the Hamiltonian of the XXZ spin chain with anisotropy parameter $\Delta=-1/2$, and various boundary conditions, possesses a special ground-state eigenvector which exhibits remarkable connections to the combinatorics of alternating sign matrices and plane partitions. The focus of the submitted paper is the XXZ spin chain with $\Delta=-1/2$ and open boundary conditions containing a single parameter. The special eigenvector for this case was recently characterized in [12], using methods involving the boundary quantum Knizhnik-Zamolodchikov equations for a multivariate generalization of the eigenvector, and several combinatorial and other properties were proved or conjectured. For example, the sum of entries of the eigenvector was conjectured to provide a generating function for totally symmetric alternating sign matrices [12, Conjecture 5.5], and an explicit expression which leads to the logarithmic bipartite fidelity (LBF) was conjectured [12, Conjecture 2.1 with $m=2$]. The LBF is a quantity which indicates the extent to which the ground-state eigenvector for a chain of length $N_1+N_2$ can be regarded as a tensor product of the ground-state eigenvectors for chains of lengths $N_1$ and $N_2$. In the submitted paper, the results and methods of [12] are used to prove the conjectured expression for the LBF in this case (Eqs. (1.6), (1.11) and (1.12)). The exact expression for finite $N_1$ and $N_2$ is then used to obtain an asymptotic expression for $N_1,N_2\to\infty$, and it is shown that this agrees with conformal field theory predictions.

I believe that this is a very interesting paper, which provides important and valuable new exact and asymptotic results for the XXZ spin chain at $\Delta=-1/2$. The exposition is very clear and comprehensive, and I have identified only a few minor matters for consideration below. Accordingly, I strongly recommend publication.

p. 7, two lines after Eq.(2.12): I think that rather than being first obtained in (44) of [33], the general solution to (2.12) for the six-vertex model was first obtained independently in (15) of https://doi.org/10.1088/0305-4470/26/12/007 and (5.12) of https://doi.org/10.1142/S0217751X94001552 The special diagonal case in (2.11) was obtained previously in Theorem 2 of https://doi.org/10.1007/BF01038545

p. 8, Eq. (2.15): Perhaps Proposition 3.1 of [12] should be cited here.

p. 8, Eqs. (2.18) and (2.21): Setting $z_2=q^{-1}z_1$ in (2.18) gives $|\Psi_2\rangle=[\beta z_1] |\downarrow\uparrow\rangle+|\uparrow\downarrow\rangle)$ rather than $[\beta z_1](|\downarrow\uparrow\rangle-|\uparrow\downarrow\rangle)$ as in (2.21). Also, the notation $|s\rangle$ might be confusing and should perhaps be changed, since $s$ was introduced previously on p. 8 for something different (i.e., a parameter with $s^2=q^3$).

p. 13, Lemma 2.7: "a most" $\rightarrow$ "at most"

p. 16, second-last line: "(the upper bound for the)'' $\rightarrow$ "(the upper bound for)''

p. 20, two lines above Corollary 3.5: Since [41] is a long paper, it may be helpful to refer to the relevant parts. In particular, the result in Corollary 3.5 follows from (3.30) of [41] (with $\mu=0$ and $\mu=1$), and references for the proof of (3.30) are given in the subsequent paragraph of [41].

p. 21, Eqs. (3.30)-(3.33): I think it should be possible to obtain an explicit expression for the determinant in (3.30) which is analogous to the expression given by (3.32) and (3.33) for the determinant in (3.31) (but it is not necessary for any of this to be done in the submitted paper). Specifically, I think that an expression for the determinant in (3.30) is
\[\prod_{i=0}^{n-2}\frac{(2i+1)\,(6i+2)!}{(6i+1)\,(2n+2i-2)!}\\
\times\sum_{0\le i\le j\le2n-2}\frac{\bigl((2n-1)(2n-i-2)+i^2\bigr)\,(2n+i-3)!\,(4n-i-4)!}{(4n-4)!\,i!\,(2n-i-1)!}\:(-1)^{i+j}\,x^j\]
which provides a counterpart to the expression
\[\prod_{i=0}^{n-2}\frac{(6i+4)!}{(2n+2i)!}\:\sum_{0\le i\le j\le2n-2}\frac{(2n+i-1)!\,(4n-i-2)!}{(4n-2)!\,i!\,(2n-i-1)!}\:(-1)^{i+j}\,x^j\]
for the determinant in (3.31).

p. 26, two lines above Eq. (4.31): "plausible assume'' $\rightarrow$ "plausible to assume''

---

## Round 2 · Author Response

Dear Editor,

We enclose our revised manuscript. Moreover, we provide here below a detailed response to the referee reports and a complete list of changes.

Sincerely yours, Gilles Parez and Christian Hagendorf.

Reply to referee 1:

We thank you for your positive review, interesting comments and questions. Here are our answers to your comments, questions and requested changes:

1) Below (1.8) (and below (2.43)), we replaced "definite magnetisation" with "magnetisation".

2) We now explicitly mention in the paragraph above (1.9) that the expressions for the overlaps of Theorem 1.1 were conjectured in [12].

3) We slightly modified the statement of Theorem 1.2 to exclude the cases where $\xi$ converges to 0 or 1.

4) Below (2.3), we now point out that our definition of the overlap differs from the standard Hermitian scalar product of quantum mechanics.

As for the meaning of the indices in (2.10), we explain that they correspond to the standard tensor-leg notation in the paragraph above (2.4). To stress that (2.4) is merely an example of this notation, we added "For example, ..." above the equation.

5) Below (2.13), we now mention that the components of $|\Psi_N\rangle$ have multiple contour-integral representations. Moreover, below (2.15), we added a sentence explaining that the simple factorised expression follows from the contour-integral representation of this special component.

6) Fixed.

7) The methods of reference [37] (previously [35]) can indeed be used. We checked explicitly that they lead to determinant formulas for the specialised symplectic characters that differ from ours (and turn out to be slightly more complicated).

However, one can prove the equality of these determinant formulas and ours' by establishing that the corresponding matrices are related one to another through conjugation with simple (and explicitly computable) triangular matrices.

We decided not to include this (technical) comment in the revised version of the text. Nonetheless, we thank the referee for their question.

8) The differential equations of Proposition 4.1 are linear with coefficients that depend polynomially on $n$. It is this polynomial dependence that forbids terms of the type $N^{-1/2}$ in the asymptotic expansion of the ratio (4.1) as $N\to \infty$ and, hence, leads to $\tau_1(z)=\bar \tau_1(z)=0$, as stated in Lemma 4.2. By recurrence, it also leads to the absence of terms of the type $N^{-(k+1/2)}$ for any integer $k\geqslant 1$.

Moreover, we believe that the polynomial dependence of the coefficients on $n$ also forbids the appearance of terms involving $\ln N$ in the asymptotic series of the ratio (4.1). Terms of this type would certainly be necessary to obtain a non-zero $g(\zeta)$. Hence, one should be able to conclude that $g(\xi)=0$ on the sole basis of the differential equations.

More generally, a detailed analysis of the differential equations should lead to the asymptotic series (4.2) and even determine all terms beyond the orders presented in (4.2). We have, however, not undertaken this analysis because the results of [30] are sufficient for our purposes.

As the reason for the different points raised by the referee is an analytic feature and not a "symmetry reason", we have refrained from modifying or expanding our manuscript with regard to this point.

9) Fixed.

Reply to referee 2:

Thank you for your review. We are pleased that you found our paper interesting and pleasant to read. As for your comments:

1) The fact that the ground-state vector of the spin chain is obtained as the homogeneous limit of the bqKZ equation is clearly stated at the beginning of section 3.2 (with a reference to [12] where it is proven). The relation between the two vectors is explicitly written in (3.27).

2) We have not investigated the possibility of deriving (3.14) with the help of Jacobi–Trudi or the Giambelli identity for the symplectic characters.

Nonetheless, we point out that there there is an alternative method for deriving (3.14), relying on the methods of reference [37] (previously [35]), as discussed above in our response to point 7 of referee 1.

Reply to referee 3:

We thank you for your positive review. Your interesting and constructive comments helped us improve our manuscript. Here are our answers to your comments, questions and requested changes:

1) Page 7: Thank you for bringing the references to several articles on the boundary Yang-Baxter equation to our attention. We added them as references [33]-[35] to the bibliography (and removed our original reference [33]).

2) Page 8: We added a sentence below (2.15). It explains that the simple factorised expressions follow from the components' multiple contour-integral expressions and cites [12]. See our response to point 5 of referee 1.

3) Page 8, (2.18): Thank you for your comment! It made us notice a missing minus sign in the expression for component $(\Psi_2)_{\uparrow\downarrow}$, which we added in the revised version of the manuscript.

Moreover, we replaced $|s\rangle$ with $|\zeta>$ throughout the text to avoid any confusion with the parameter $s$ obeying $s^2=q^3$.

4) Page 14 (previously page 13): Fixed.

5) Page 17 (previously page 16): Fixed.

6) Page 20, above Corollary 3.5: We added an explicit reference to Theorem 37 with $\mu=0$ and $\mu=1$ of [43] (previously [41]).

7) Page 21, (3.30)-(3.33): We thank the referee for their interesting observation.

We have compared for small values of $n$ the referee's explicit expression for the determinant in the case of even $N_1, N_2$ and our determinant in (3.30). The two expressions do indeed coincide, provided that one replaces $n$ with $n+1$ in the referee's expression.

Nonetheless, as suggested by the referee, we decided not to elaborate on this point in the revised version of the manuscript.

8) Page 26: Fixed.

---

## Round 2 · List of Changes

• Page 3, below (1.8): Changed "definite magnetisation" to "magnetisation" (R1 point 1). The same replacement was made at the top of page 12.

  • Page 3, above (1.9): Added a sentence explaining that the expressions of Theorem 1 were conjectured in [12] (R1 point 2).

  • Page 4, Theorem 1.2: Modified statement of the theorem to exclude the cases where $\xi$ converges to $0$ or $1$ (R1 point 3).

  • Page 6, below (2.3): Modified the paragraph to point out that the overlap is different from the standard Hermitian scalar product of quantum mechanics (R1 point 4).

  • Page 6, above (2.4): Replaced "For $M=1$" with "For example, if $M=1$" (R1 point 4).

  • Page 7, below (2.10): Replaced "the following $K$-matrix" with "a diagonal $K$-matrix".

  • Page 7, below (2.12): Added references suggested by referee 3 (R3 point 1).

  • Page 7, below (2.13): Added a comment explaining that the components of $|\Psi_N\rangle$ that do not vanish trivially have explicit expressions in terms of multiple contour integrals (related to R1 point 5 and R3 point 2).

  • Page 8, below (2.15): Added a sentence explaining that the factorised expression follows from the multiple contour-integral formula for the component and cited [12] (R1 point 5, R3 point 2).

  • Page 8, (2.18): Added missing minus sign in the expression of $(\Psi_2)_{\uparrow\downarrow}$ (R3 point 3).

  • Page 9, just above (2.21): Replaced $|s\rangle$ with $|\zeta\rangle$ (R3 point 3). The same replacement was made on page 8, (2.21); page 9, first paragraph and (2.22); and page 10, first paragraph.

  • Page 14, Lemma 2.7: Corrected "is a most" to "is at most" (R3 point 4).

  • Page 17, above (2.76): Replaced "(the upper bound for the) the degree width" with "(the upper bound for) the degree width" (R3 point 5).

  • Page 18, below (3.7): Replaced "following the proof on Andreev's formula" with "using Andreev's formula" (R1 point 6).

  • Page 20, above Corollary 3.5: Replaced "with the help of Krattenthaler's formula [41]" with the detailed reference "[43, Theorem 37 with $\mu=0$ and $\mu=1$]" (R3 point 6).

  • Page 24, below (4.21): Replaced "parametrisation" with "parameterisation" to be consistent throughout the manuscript.

  • Page 26, above (4.31): Corrected "is plausible assume" to "is plausible to assume" (R1 point 9, R3 point 8).

  • Page 29: Added to [12] a reference to a published erratum to the original article.

  • Page 30: Replaced original reference [33] with new references [33]-[35] suggested by referee 3 (R3 point 1).

---

## Editorial Decision

published